# NEURAL OPTIMAL TRANSPORT WITH GENERAL COST FUNCTIONALS

## ABSTRACT

We present a novel neural-networks-based algorithm to compute optimal transport (OT) plans and maps for general cost functionals. The algorithm is based on a saddle point reformulation of the OT problem and generalizes prior OT methods for weak and strong cost functionals. As an application, we construct a functional to map data distributions with preserving the class-wise structure of data.

## 1 INTRODUCTION

Optimal transport (OT) is a powerful framework to solve mass-moving and generative modeling problems for data distributions. Recent works (Korotin et al., 2022c; Rout et al., 2022; Korotin et al., 2021b; Fan et al., 2021a; Daniels et al., 2021) propose scalable neural methods to compute **OT plans** (or maps). They show that the learned transport plan (or map) can be used directly as the generative model in data synthesis (Rout et al., 2022) and unpaired learning (Korotin et al., 2022c; Rout et al., 2022; Daniels et al., 2021; Gazdieva et al., 2022). Compared to WGANs (Arjovsky et al., 2017) which employ **OT cost** as the loss for generator (Rout et al., 2022, §3), these methods provide better flexibility: the properties of the learned model can be controlled by the transport cost function.

Existing neural OT plan (or map) methods consider distance-based cost functions, e.g., weak or strong quadratic costs (Korotin et al., 2022c; Fan et al., 2021a; Gazdieva et al., 2022). Such costs are suitable for the tasks of unpaired image-to-image style translation (Zhu et al., 2017, Figures 1,2) and image restoration (Lugmayr et al., 2020). However, they do not take into account the class-wise structure of data or available side information, e.g., the class labels. As a result, such costs are hardly applicable to certain tasks such as the dataset transfer where the preservation the class-wise structure is needed (Figure 1). We tackle this issue.

**Contributions.** We propose the extension of neural OT which allows to apply it to previously unreleased problems. For this, we develop a novel neural-networks-based algorithm to compute optimal transport plans for *general cost functionals* (§3). As an example, we construct (§4) and test (§6) the functional for mapping data distributions with the preservation the class-wise structure.

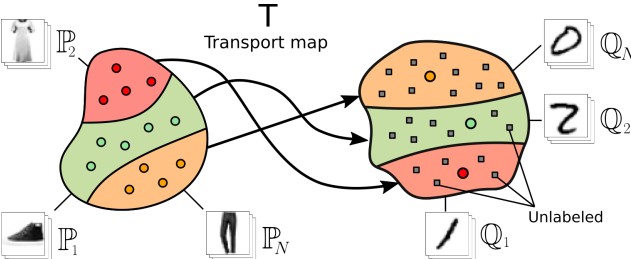

Figure 1: The setup of *class-guided dataset transfer*. Input $\mathbb{P} = \sum_n \alpha_n \mathbb{P}_n$, target $\mathbb{Q} = \sum_n \beta_n \mathbb{Q}_n$ distributions are mixtures of $N$ classes. The task is to learn a transport map $T$ preserving the class. The learner has the access to labeled input data $\sim \mathbb{P}$ and only *partially labeled* target data $\sim \mathbb{Q}$.

**Notation.** The notation of our paper is based on that of (Paty & Cuturi, 2020; Korotin et al., 2022c). For a compact Hausdorf space $\mathcal{S}$ we use $\mathcal{P}(\mathcal{S})$ to denote the set of Borel probability distributions on $\mathcal{S}$. We denote the space of continuous $\mathbb{R}$-valued functions on $\mathcal{S}$ endowed with the supremum norm by $\mathcal{C}(\mathcal{S})$. Its dual space is the space $\mathcal{M}(\mathcal{S}) \supset \mathcal{P}(\mathcal{S})$ of finite signed Borel measures over $\mathcal{S}$. For a

functional $\mathcal{F} \colon \mathcal{M}(\mathcal{S}) \to \mathbb{R} \cup \{\infty\}$, we use $\mathcal{F}^*(h) \stackrel{def}{=} \sup_{\pi \in \mathcal{M}(\mathcal{S})} \left[ \int_{\mathcal{S}} h(s) d\pi(s) - \mathcal{F}(\pi) \right]$ to denote its convex conjugate functional $\mathcal{F}^* : C(\mathcal{S}) \to \mathbb{R} \cup \{\infty\}$.

Let $\mathcal{X}, \mathcal{Y}$ be compact Hausdorf spaces and $\mathbb{P} \in \mathcal{P}(\mathcal{X})$, $\mathbb{Q} \in \mathcal{P}(\mathcal{Y})$. We use $\Pi(\mathbb{P}) \subset \mathcal{P}(\mathcal{X} \times \mathcal{Y})$ to denote the subset of probability measures on $\mathcal{X} \times \mathcal{Y}$ which projection onto the first marginal is $\mathbb{P}$. We use $\Pi(\mathbb{P}, \mathbb{Q}) \subset \Pi(\mathbb{P})$ to denote the subset of probability measures (transport plans) on $\mathcal{X} \times \mathcal{Y}$ with marginals $\mathbb{P}, \mathbb{Q}$. For $u, v \in \mathcal{C}(\mathcal{X}), \mathcal{C}(\mathcal{Y})$ we write $u \oplus v \in \mathcal{C}(\mathcal{X} \times \mathcal{Y})$ to denote the function $u \oplus v : (x, y) \mapsto u(x) + v(y)$. For a functional $\mathcal{F} : \mathcal{M}(\mathcal{X} \times \mathcal{Y}) \to \mathbb{R}$ we say that it is separably *-increasing if for all functions $u, v \in \mathcal{C}(\mathcal{X}), \mathcal{C}(\mathcal{Y})$ and any function $c \in \mathcal{C}(\mathcal{X} \times \mathcal{Y})$ from $u \oplus v \leq c$ (point-wise) it follows $\mathcal{F}^*(u \oplus v) \leq \mathcal{F}^*(c)$. For a measurable map $T : \mathcal{X} \times \mathcal{Z} \to \mathcal{Y}$, we denote the associated push-forward operator by $T_\#$. For $\mathbb{Q}_1, \mathbb{Q}_2 \in \mathcal{P}(\mathcal{Y})$ with $\mathcal{Y} \subset \mathbb{R}^D$, the (square of) energy distance $\mathcal{E}$ (Rizzo & Székely, 2016) between them is:

$$\mathcal{E}^2(\mathbb{Q}_1, \mathbb{Q}_2) = \mathbb{E}\|Y_1 - Y_2\| - \frac{1}{2}\mathbb{E}\|Y_1 - Y_1'\| - \frac{1}{2}\mathbb{E}\|Y_2 - Y_2'\|, \tag{1}$$

where $Y_1 \sim \mathbb{Q}_1, Y_1' \sim \mathbb{Q}_1, Y_2 \sim \mathbb{Q}_2, Y_2' \sim \mathbb{Q}_2$ are independent random vectors. Energy distance (1) is a particular case of the Maximum Mean Discrepancy (Sejdinovic et al., 2013). It equals zero only when $\mathbb{Q}_1 = \mathbb{Q}_2$.

## 2 PRELIMINARIES

In this section, we provide key concepts of the optimal transport theory. Thought the paper, we consider compact $\mathcal{X} = \mathcal{Y} \subset \mathbb{R}^D$ and $\mathbb{P}, \mathbb{Q} \in \mathcal{P}(\mathcal{X}), \mathcal{P}(\mathcal{Y})$.

**Strong OT.** For a cost function $c \in \mathcal{C}(\mathcal{X} \times \mathcal{Y})$, the *optimal transport cost* between $\mathbb{P}, \mathbb{Q}$ is

$$\text{Cost}(\mathbb{P}, \mathbb{Q}) \stackrel{def}{=} \inf_{\pi \in \Pi(\mathbb{P}, \mathbb{Q})} \int_{\mathcal{X} \times \mathcal{Y}} c(x, y) d\pi(x, y), \tag{2}$$

see (Villani, 2008, §1) Problem (2) admits a minimizer $\pi^* \in \Pi(\mathbb{P}, \mathbb{Q})$ which is called an *OT plan* (Santambrogio, 2015, Theorem 1.4). It may be not unique (Peyré et al., 2019, Remark 2.3). Intuitively, the cost function $c(x, y)$ measures how hard it is to move a mass piece between points $x \in \mathcal{X}$ and $y \in \mathcal{Y}$. That is, $\pi^*$ shows how to optimally distribute the mass of $\mathbb{P}$ to $\mathbb{Q}$, i.e., with the minimal effort.

For cost functions $c(x, y) = \|x - y\|$ and $c(x, y) = \frac{1}{2}\|x - y\|^2$, the OT cost (2) is called the Wasserstein-1 ($\mathbb{W}_1$) and the (square of) Wasserstein-2 ($\mathbb{W}_2$) distance, respectively, see (Villani, 2008, §1) or (Santambrogio, 2015, §1, 2).

**Weak OT.** Consider a *weak* cost function $C : \mathcal{X} \times \mathcal{P}(\mathcal{Y}) \to \mathbb{R}$. Its inputs are a point $x \in \mathcal{X}$ and a distribution of $y \in \mathcal{Y}$. The weak OT cost is (Gozlan et al., 2017; Backhoff-Veraguas et al., 2019)

$$\text{Cost}(\mathbb{P}, \mathbb{Q}) \stackrel{def}{=} \inf_{\pi \in \Pi(\mathbb{P}, \mathbb{Q})} \int_{\mathcal{X}} C\big(x, \pi(\cdot|x)\big) d\pi(x), \tag{3}$$

where $\pi(\cdot|x)$ denotes the conditional distribution. Weak formulation (3) reduces to strong formulation (2) when $C(x, \mu) = \int_{\mathcal{Y}} c(x, y) d\mu(y)$. An other example of a weak cost function is the $\gamma$-weak quadratic cost $C(x, \mu) = \int_{\mathcal{Y}} \frac{1}{2}\|x - y\|^2 d\mu(y) - \frac{\gamma}{2}\text{Var}(\mu)$, where $\gamma \geq 0$ and $\text{Var}(\mu)$ is the variance of $\mu$, see (Korotin et al., 2022c, Eq. 5), (Alibert et al., 2019, §5.2), (Gozlan & Juillet, 2020, §5.2) for details. For this cost, we denote the optimal value of (3) by $\mathcal{W}_{2,\gamma}^2$ and call it $\gamma$-*weak Wasserstein-2*.

**Regularized OT.** The expression inside (2) is a linear functional. It is common to add a lower-semi-continuous convex regularizer $\mathcal{R} : \mathcal{M}(\mathcal{X} \times \mathcal{Y}) \to \mathbb{R} \cup \{\infty\}$ with weight $\gamma > 0$:

$$\text{Cost}(\mathbb{P}, \mathbb{Q}) \stackrel{def}{=} \inf_{\pi \in \Pi(\mathbb{P}, \mathbb{Q})} \left\{ \int_{\mathcal{X} \times \mathcal{Y}} c(x, y) d\pi(x, y) + \gamma \mathcal{R}(\pi) \right\}. \tag{4}$$

Regularized formulation (4) typically provides several advantages over original formulation (2). For example, if $\mathcal{R}(\pi)$ is strictly convex, the expression inside (4) is a strictly convex functional in $\pi$ and yields the unique OT plan $\pi^*$. Besides, regularized OT typically has better sample complexity (Genevay, 2019; Mena & Niles-Weed, 2019; Genevay et al., 2019). Common regularizers are the entropic (Cuturi, 2013b), quadratic (Essid & Solomon, 2018), Lasso (Courty et al., 2016), etc.

**General OT.** Let $\mathcal{F} : \mathcal{M}(\mathcal{X} \times \mathcal{Y}) \to \mathbb{R} \cup \{+\infty\}$ be a convex lower-semi-continuous functional. Assume that there exists $\pi \in \Pi(\mathbb{P}, \mathbb{Q})$ for which $\mathcal{F}(\pi) < \infty$. Consider the problem:

$$\text{Cost}(\mathbb{P}, \mathbb{Q}) \stackrel{\text{def}}{=} \inf_{\pi \in \Pi(\mathbb{P}, \mathbb{Q})} \mathcal{F}(\pi). \tag{5}$$

The problem is a *generalization* of strong OT (2), weak OT (3), regularized OT (4); following (Paty & Cuturi, 2020), we call problem (5) a general OT problem. Surprisingly, regularized OT (4) represents the same problem: it is enough to put $c(x, y) \equiv 0$, $\gamma = 1$ and $\mathcal{R}(\pi) = \mathcal{F}(\pi)$ to obtain (5) from (4). That is, regularized OT (4) and general OT (5) can be viewed as equivalent formulations.

**Existence and duality.** With mild assumptions on $\mathcal{F}$, problem (5) admits a minimizer $\pi^*$ (Paty & Cuturi, 2020, Lemma 1). If $\mathcal{F}$ is separately *-increasing, the dual problem is

$$\text{Cost}(\mathbb{P}, \mathbb{Q}) = \sup_{u,v} \left[ \int_{\mathcal{X}} u(x) d\mathbb{P}(x) + \int_{\mathcal{Y}} v(y) d\mathbb{Q}(y) - \mathcal{F}^*(u \oplus v) \right], \tag{6}$$

where optimization is performed over $u, v \in \mathcal{C}(\mathcal{X}), \mathcal{C}(\mathcal{Y})$ which are called *potentials*, see (Paty & Cuturi, 2020, Theorem 2) for details. Problem (6) also admits a pair of minimizers $u^*, v^*$. The popular regularized functionals (4) are indeed separately *-increasing, including the OT regularized with entropy (Paty & Cuturi, 2020, Example 7) or $L^p$ (Paty & Cuturi, 2020, Example 8). That is, formulation (6) subsumes many known duality formulas for regularized OT.

## 3 ALGORITHM FOR LEARNING OPTIMAL TRANSPORT PLANS

In this section, we derive an algorithm to solve general OT problem (5) with neural networks. We prove that (5) can be reformulated as a saddle point optimization problem (§3.1) from the solution of which one may implicitly recover the OT plan $\pi^*$ (§3.2). We give the proofs in Appendix A.

### 3.1 MAXIMIN REFORMULATION OF THE DUAL PROBLEM

In this subsection, we derive the dual form, which is alternative to (6) and can be used to get the OT plan $\pi^*$. Our two following theorems constitute the main theoretical idea of our approach.

**Theorem 1** (Maximin reformulation of the dual problem). *For *-separately increasing convex and lower-semi-continuous functional $\mathcal{F} : \mathcal{M}(\mathcal{X} \times \mathcal{Y}) \to \mathbb{R} \cup \{+\infty\}$ it holds*

$$\text{Cost}(\mathbb{P}, \mathbb{Q}) = \sup_v \inf_{\pi \in \Pi(\mathbb{P})} \mathcal{L}(v, \pi) = \sup_v \inf_{\pi \in \Pi(\mathbb{P})} \left\{ [\mathcal{F}(\pi) - \int_{\mathcal{Y}} v(y) d\pi(y)] + \int_{\mathcal{Y}} v(y) d\mathbb{Q}(y) \right\}, \tag{7}$$

*where the* sup *is taken over $v \in \mathcal{C}(\mathcal{Y})$ and $\pi(y)$ is the marginal distribution over $y$ of the plan $\pi$.* From (7) we also see that it is enough to consider values of $\mathcal{F}$ in $\pi \in \Pi(\mathbb{P}) \subset \mathcal{M}(\mathcal{X} \times \mathcal{Y})$. For convention, in further derivations we always consider $\mathcal{F}(\pi) = +\infty$ for $\pi \in \mathcal{M}(\mathcal{X} \times \mathcal{Y}) \setminus \Pi(\mathbb{P})$.

**Theorem 2** (Optimal saddle points provide optimal plans). *Let $v^* \in \arg \sup_v \inf_{\pi \in \Pi(\mathbb{P})} \mathcal{L}(v, \pi)$ be any optimal potential. Then for every optimal transport plan $\pi^* \in \Pi(\mathbb{P}, \mathbb{Q})$ it holds:*

$$\pi^* \in \arg \inf_{\pi \in \Pi(\mathbb{P})} \mathcal{L}(v^*, \pi). \tag{8}$$

If $\mathcal{F}$ is strictly convex in $\pi \in \Pi(\mathbb{P})$, then $\mathcal{L}(v^*, \pi)$ is strictly convex as a functional of $\pi$. Consequently, it has a *unique* minimizer. As a result, expression (8) is an equality. We have the following corollary.

**Corollary 1** (Every optimal saddle point provides the optimal transport plan). *Assume additionally that $\mathcal{F}$ is strictly convex. Then the unique OT plan satisfies $\pi^* = \arg \inf_{\pi \in \Pi(\mathbb{P})} \mathcal{L}(v^*, \pi)$.*

Thanks to our results above, one may solve (7), obtain the OT plan $\pi^*$ from the solution $(v^*, \pi^*)$ of the saddle point problem (7). We propose an algorithm to do this in the next subsections.

### 3.2 REPLACING MEASURES WITH STOCHASTIC MAPS

Formulation (7) requires optimization over probability measures $\pi \in \Pi(\mathbb{P})$. To make it practically feasible, we reformulate it as the optimization over functions $T$ which generate these measures $\pi$.

We introduce a latent space $\mathcal{Z} = \mathbb{R}^Z$ and an atomless measure $\mathbb{S} \in \mathcal{P}(\mathcal{Z})$ on it, e.g., $\mathbb{S} = \mathcal{N}(0, I_Z)$. For every $\pi \in \mathcal{P}(\mathcal{X} \times \mathcal{Y})$, there exists a measurable function $T = T_\pi : \mathcal{X} \times \mathcal{Z} \to \mathcal{Y}$ which implicitly represents it (Korotin et al., 2022c, §4.1). Such $T_\pi$ satisfies $T_\pi(x, \cdot) \sharp \mathbb{S} = \pi(y|x)$ for all

---

**Algorithm 1:** Neural optimal transport (NOT) for general cost functionals

---

**Input**    : distributions $\mathbb{P}, \mathbb{Q}, \mathbb{S}$ accessible by samples;
         mapping network $T_\theta : \mathbb{R}^P \times \mathbb{R}^S \to \mathbb{R}^Q$; potential network $v_\omega : \mathbb{R}^Q \to \mathbb{R}$;
         number of inner iterations $K_T$; empirical estimator $\widehat{\mathcal{F}}\big(X, T(X, Z)\big)$ for cost $\widetilde{\mathcal{F}}(T)$;

**Output :** learned stochastic OT map $T_\theta$ representing an OT plan between distributions $\mathbb{P}, \mathbb{Q}$;

**repeat**

     Sample batches $Y \sim \mathbb{Q}$, $X \sim \mathbb{P}$ and for each $x \in X$ sample batch $Z[x] \sim \mathbb{S}$;

     $\mathcal{L}_v \leftarrow \frac{1}{|X|} \sum\limits_{x \in X} \frac{1}{|Z[x]|} \sum\limits_{z \in Z[x]} v_\omega\big(T_\theta(x,z)\big) - \frac{1}{|Y|} \sum\limits_{y \in Y} v_\omega(y)$;

     Update $\omega$ by using $\frac{\partial \mathcal{L}_v}{\partial \omega}$;

     **for** $k_T = 1, 2, \ldots, K_T$ **do**

         Sample batch $X \sim \mathbb{P}$ and for each $x \in X$ sample batch $Z[x] \sim \mathbb{S}$;

         $\mathcal{L}_T \leftarrow \widehat{\mathcal{F}}\big(X, T_\theta(X, Z)\big) - \frac{1}{|X|} \sum\limits_{x \in X} \frac{1}{|Z[x]|} \sum\limits_{z \in Z[x]} v_\omega\big(T_\theta(x,z)\big)$;

         Update $\theta$ by using $\frac{\partial \mathcal{L}_T}{\partial \theta}$;

**until** *not converged*;

---

$x \in \mathcal{X}$. That is, given $x \in \mathcal{X}$ and a random latent vector $z \sim \mathbb{S}$, the function $T$ produces sample $T_\pi(x, z) \sim \pi(y|x)$. In particular, if $x \sim \mathbb{P}$, the random vector $[x, T_\pi(x, z)]$ is distributed as $\pi$. Thus, every $\pi \in \Pi(\mathbb{P})$ can be implicitly represented as a function $T_\pi : \mathcal{X} \times \mathcal{Z} \to \mathcal{Y}$. Note that, in general, there might exist *several* suitable functions $T_\pi$.

Every measurable function $T : \mathcal{X} \times \mathcal{Z} \to \mathcal{Y}$ is an implicit representation of the measure $\pi_T$ which is the joint distribution a random vector $[x, T(x, z)]$ with $x \sim \mathbb{P}, z \sim \mathbb{S}$. Consequently, the optimization over $\pi \in \Pi(\mathbb{P})$ is *equivalent* to the optimization over measurable functions $T : \mathcal{X} \times \mathcal{Z} \to \mathcal{Y}$. From our Theorem 1, we have the following corollary.

**Corollary 2.** *For \*-separately increasing, lower-semi-continuous and convex $\mathcal{F}$ it holds*

$$\text{Cost}(\mathbb{P}, \mathbb{Q}) = \sup_v \inf_T \widetilde{\mathcal{L}}(v, T) = \sup_v \inf_T \left\{ \widetilde{\mathcal{F}}(T) - \int_{\mathcal{X} \times \mathcal{Z}} v\big(T(x, z)\big) d\mathbb{P}(x) d\mathbb{S}(z) + \int_{\mathcal{Y}} v(y) d\mathbb{Q}(y) \right\}, \quad (9)$$

*where the* $\sup$ *is taken over potentials* $v \in \mathcal{C}(\mathcal{Y})$ *and* $\inf$ *– over measurable functions* $T : \mathcal{X} \times \mathcal{Z} \to \mathcal{Y}$. *Here we identify* $\widetilde{\mathcal{F}}(T) \overset{def}{=} \mathcal{F}(\pi_T)$ *and* $\widetilde{\mathcal{L}}(v, T) \overset{def}{=} \mathcal{L}(v, \pi_T)$.

Following the notation of (Korotin et al., 2022c), we say that functions $T$ are stochastic maps (Figure 6). We say that $T^*$ is a *stochastic OT map* if it represents some OT plan $\pi^*$, i.e., $T^*(x, \cdot)\sharp\mathbb{S} = \pi^*(\cdot|x)$ holds $\mathbb{P}$-almost surely for all $x \in \mathcal{X}$. From Theorem 2 and Corollary 1, we obtain the following result.

**Corollary 3** (Optimal saddle points provide stochastic OT maps)**.** *Let* $v^* \in \arg \sup_v \inf_T \widetilde{\mathcal{L}}(v, T)$ *be any optimal potential. Then for every stochastic OT map* $T^*$ *it holds:*

$$T^* \in \arg \inf_T \widetilde{\mathcal{L}}(v^*, T). \quad (10)$$

*If $\mathcal{F}$ is strictly convex in $\pi$, we have* $T^* \in \arg \inf_T \widetilde{\mathcal{L}}(v^*, T) \Leftrightarrow T^*$ *is a stochastic OT map.*

From our results it follows that by solving (9) and obtaining an optimal saddle point $(v^*, T^*)$, one gets a stochastic OT map $T^*$. To guarantee that all the solutions are OT maps, one may consider adding strictly convex regularizers to $\mathcal{F}$ with some small weight, e.g., the conditional kernel variance (Korotin et al., 2022b). Problem (9) replaces the *constrained* optimization over measures $\pi \in \Pi(\mathbb{P})$ in (7) with optimization over stochastic maps $T$ in (9), making it practically feasible.

### 3.3   GENERIC PRACTICAL OPTIMIZATION PROCEDURE

To approach the problem (9) in practice, we use neural nets $T_\theta : \mathbb{R}^D \times \mathbb{R}^S \to \mathbb{R}^D$ and $v_\omega : \mathbb{R}^D \to \mathbb{R}$ to parametrize $T$ and $v_\omega$, respectively. We train them with the stochastic gradient ascent-descent (SGAD) by using random batches from $\mathbb{P}, \mathbb{Q}, \mathbb{S}$. The optimization procedure is given in Algorithm 1. In the implementation, to update networks $T_\theta$ and $v_\omega$, we use Adam optimizer (Kingma & Ba, 2014).

---

**Algorithm 2:** Neural optimal transport (NOT) with the class-guided cost functional $\widetilde{\mathcal{F}}_{\mathrm{G}}$.

---

**Input** : distributions $\mathbb{P} = \sum_n \alpha_n \mathbb{P}_n$, $\mathbb{Q} = \sum_n \beta_n \mathbb{Q}_n$, $\mathbb{S}$ accessible by samples (*unlabeled*);
  weights $\alpha_n$ are known and samples from each $\mathbb{P}_n, \mathbb{Q}_n$ are accessible (*labeled*);
  mapping network $T_\theta : \mathbb{R}^P \times \mathbb{R}^S \to \mathbb{R}^Q$; potential network $v_\omega : \mathbb{R}^Q \to \mathbb{R}$;
  number of inner iterations $K_T$;

**Output** : learned stochastic OT map $T_\theta$ representing an OT plan between distributions $\mathbb{P}, \mathbb{Q}$;

**repeat**

   | Sample (unlabeled) batches $Y \sim \mathbb{Q}$, $X \sim \mathbb{P}$ and for each $x \in X$ sample batch $Z[x] \sim \mathbb{S}$;

   | $\mathcal{L}_v \leftarrow \frac{1}{|X|} \sum_{x \in X} \frac{1}{|Z[x]|} \sum_{z \in Z[x]} v_\omega\big(T_\theta(x, z)\big) - \frac{1}{|Y|} \sum_{y \in Y} v_\omega(y)$;

   | Update $\omega$ by using $\frac{\partial \mathcal{L}_v}{\partial \omega}$;

   | **for** $k_T = 1, 2, \ldots, K_T$ **do**

      | Pick $n \in \{1, 2, \ldots, N\}$ at random with probabilities $(\alpha_1, \ldots, \alpha_N)$;

      | Sample (labeled) batches $X_n \sim \mathbb{P}_n$, $Y_n \sim \mathbb{Q}$; for each $x \in X$ sample batch $Z_n[x] \sim \mathbb{S}$;

      | $\mathcal{L}_T \leftarrow \widehat{\Delta \mathcal{E}^2}\big(X_n, T(X_n, Z_n), Y_n\big) - \frac{1}{|X_n|} \sum_{x \in X_n} \frac{1}{|Z_n[x]|} \sum_{z \in Z_n[x]} v_\omega\big(T_\theta(x, z)\big)$;

      | Update $\theta$ by using $\frac{\partial \mathcal{L}_T}{\partial \theta}$;

**until** *not converged*;

---

Our Algorithm 1 requires an empirical estimator $\widehat{\mathcal{F}}$ for $\widetilde{\mathcal{F}}(T)$. Providing such an estimator might be non-trivial for general $\mathcal{F}$. If $\mathcal{F}(\pi) = \int_{\mathcal{X}} C(x, \pi(\cdot|x)) d\mathbb{P}(x)$, i.e., the cost is weak (3), one may use the following *unbiased* Monte-Carlo estimator: $\widehat{\mathcal{F}}\big(X, T(X, Z)\big) \stackrel{def}{=} |X|^{-1} \sum_{x \in X} \widehat{C}\big(x, T(x, Z[x])\big)$, where $\widehat{C}$ is the respective estimator for the weak cost $C$ and $Z[x]$ denotes a random batch of latent vectors $z \sim \mathbb{S}$ for a given $x \in \mathcal{X}$. For strong costs and the $\gamma$-weak quadratic cost, the estimator $\widehat{C}$ is given by (Korotin et al., 2022c, Eq. 22 and 23) and our Algorithm 1 for general OT 5 reduces to the NOT algorithm (Korotin et al., 2022c, Algorithm 1) for weak (3) or strong (2) OT. Unlike the predecessor, our algorithm is suitable for general OT formulation (5). In the next section, we propose a cost functional $\mathcal{F}_{\mathrm{G}}$ (and provide an estimator for it) to solve the class-guided dataset transfer task.

## 4    CLASS-GUIDED DATASET TRANSFER WITH NEURAL OPTIMAL TRANSPORT

In this section, we show that general cost functionals (5) are useful, for example, for the class-guided dataset transfer. To begin with, we theoretically formalize the problem setup.

Let each input $\mathbb{P}$ and output $\mathbb{Q}$ distributions be a mixture of $N$ distributions (*classes*) $\{\mathbb{P}_n\}_{n=1}^N$ and $\{\mathbb{Q}_n\}_{n=1}^N$, respectively. That is $\mathbb{P} = \sum_{n=1}^N \alpha_n \mathbb{P}_n$ and $\mathbb{Q} = \sum_{n=1}^N \beta_n \mathbb{Q}_n$ where $\alpha_n, \beta_n \geq 0$ are the respective weights (*class prior probabilities*) satisfying $\sum_{n=1}^N \alpha_n = 1$ and $\sum_{n=1}^N \beta_n = 1$. In this general setup, we aim to find the transport plan $\pi(x, y) \in \Pi(\mathbb{P}, \mathbb{Q})$ for which the classes of $x \in \mathcal{X}$ and $y \in \mathcal{Y}$ are the same for as much pairs $(x, y) \sim \pi$ as possible. That is, its respective stochastic map $T$ should map each component $\mathbb{P}_n$ (class) of $\mathbb{P}$ to the respective component $\mathbb{Q}_n$ (class) of $\mathbb{Q}$.

The task above is related to *domain adaptation* or *transfer learning* problems. It does not always have a solution with each $\mathbb{P}_n$ exactly mapped to $\mathbb{Q}_n$ due to possible prior/posterior shift (Kouw & Loog, 2018). We aim to find a stochastic map $T$ between $\mathbb{P}$ and $\mathbb{Q}$ satisfying $T_\sharp(\mathbb{P}_n \times \mathbb{S}) \approx \mathbb{Q}_n$ for all $n = 1, \ldots, N$. To solve the above discussed problem, we propose the following functional:

$$\mathcal{F}_{\mathrm{G}}(\pi) \stackrel{def}{=} \widetilde{\mathcal{F}}_{\mathrm{G}}(T_\pi) \stackrel{def}{=} \sum_{n=1}^N \alpha_n \mathcal{E}^2\big(T_\pi \sharp (\mathbb{P}_n \times \mathbb{S}), \mathbb{Q}_n\big), \tag{11}$$

where $\mathcal{E}$ denotes the energy distance (1). Functional (11) is non-negative and attains zero value when the components of $\mathbb{P}$ are correctly mapped to the respective components of $\mathbb{Q}$ (if this is possible).

**Theorem 3.** *Functional $\mathcal{F}_{\mathrm{G}}(\pi)$ is convex[1] in $\pi \in \Pi(\mathbb{P})$ and $*$-separably increasing.*

In practice, each of the terms $\mathcal{E}^2\big(T_\pi \sharp (\mathbb{P}_n \times \mathbb{S}), \mathbb{Q}_n\big)$ in (11) admits estimation from samples from $\pi$.

---

[1]The functional $\mathcal{F}_{\mathrm{G}}(\pi)$ is not necessarily *strictly* convex.

**Proposition 1** (Estimator for $\mathcal{E}^2$). *Let $X_n \sim \mathbb{P}_n$ be a batch of $K_X$ samples from class $n$. For each $x \in X_n$ let $Z_n[x] \sim \mathbb{S}$ be a latent batch of size $K_Z$. Consider a batch $Y_n \sim \mathbb{Q}_n$ of size $K_Y$. Then*

$$\widehat{\Delta \mathcal{E}^2}\big(X_n, T(X_n, Z_n), Y_n\big) \stackrel{def}{=} \frac{1}{K_Y \cdot K_X \cdot K_Z} \sum_{y \in Y_n} \sum_{x \in X_n} \sum_{z \in Z_n[x]} \|y - T(x, z)\| -$$

$$\frac{1}{2 \cdot (K_X^2 - K_X) \cdot K_Z^2} \sum_{x \in X_n} \sum_{z \in Z_n[x]} \sum_{x' \in X_n \setminus \{x\}} \sum_{z' \in Z_{x'}} \|T(x, z) - T(x', z')\| \quad (12)$$

*is an estimator of $\mathcal{E}^2\big(T\sharp(\mathbb{P}_n \times \mathbb{S}), \mathbb{Q}_n\big)$ up to a constant $\underline{T}$-independent shift.*

To estimate $\widetilde{\mathcal{F}}_G(T)$, one may separately estimate terms $\mathcal{E}^2\big(T\sharp(\mathbb{P}_n \times \mathbb{S}), \mathbb{Q}_n\big)$ for each $n$ and sum them up with weights $\alpha_n$. We only estimate only $n$-th term with probability $\alpha_n$ at each iteration.

We highlight the **two key details** of the estimation of (11) which are significantly different from estimation weak OT costs (3) appearing in related works (Korotin et al., 2022c; Fan et al., 2021a; Korotin et al., 2021b). First, one has to sample not just from the input distribution $\mathbb{P}$, but *separately* from each its component (class) $\mathbb{P}_n$. Moreover, one also has to be able to *separately* sample from the target distribution's $\mathbb{Q}$ components $\mathbb{Q}_n$. This is the part where the *guidance* (semi-supervision) happens. We note that to estimate costs such as strong or weak (3), no target samples from $\mathbb{Q}$ are needed at all, i.e., they can be viewed as unsupervised.

In practice, we assume that the learner is given a *labeled* empirical sample from $\mathbb{P}$ for training. In contrast, we assume that the available samples from $\mathbb{Q}$ are only *partially labeled* (with $\geq 1$ labeled data point per class). That is, we know the class label only for a limited amount of data (Figure 1). In this case, all $n$ cost terms (12) can still be stochastically estimated. These cost terms are used to learn the transport map $T_\theta$ in Algorithm 1. The remaining (unlabeled) samples will be used when training the potential $f_\omega$, as labels are not needed to update the potential in (9). We provide the detailed procedure for learning with the functional $\mathcal{F}_G$ (11) in Algorithm 2.

## 5 RELATED WORK

**Neural networks for OT.** To the best of our knowledge, our method (§3) is the *first* to compute **OT plans** for general cost functionals (5). Our duality formula (9) *subsumes* previously known formulas for *weak* (3) and *strong* (2) functionals (Korotin et al., 2022a, Eq. 7), (Korotin et al., 2021b, Eq. 9), (Rout et al., 2022, Eq. 14), (Fan et al., 2021a, Eq. 11), (Henry-Labordere, 2019, Eq. 11), (Gazdieva et al., 2022, Eq. 10). For the strong quadratic cost, (Makkuva et al., 2019), (Korotin et al., 2021a, Eq. 10) consider analogous to (9) formulations *restricted* to convex potentials; they use Input Convex Neural Networks (Amos et al., 2017) to approximate them. These nets are popular in OT (Korotin et al., 2021c; Mokrov et al., 2021; Fan et al., 2021a; Bunne et al., 2021; Alvarez-Melis et al., 2021) but OT algorithms based on them are outperformed (Korotin et al., 2021b) by the above-mentioned unrestricted formulations. In (Genevay et al., 2016; Seguy et al., 2017; Daniels et al., 2021; Fan et al., 2021b), the authors propose methods for $f$-divergence *regularized* functionals (4). The first two methods recover biased plans which is a notable issue in high dimensions (Korotin et al., 2021b, §4.2). Method (Daniels et al., 2021) is computationally heavy due to using the Langevin dynamics.

Many approaches in generative learning use **OT cost** as the loss function to update generative models (WGANs (Arjovsky & Bottou, 2017; Petzka et al., 2017; Liu et al., 2019)). They are *not related* to our work as they do not compute OT plans (or maps). Importantly, saddle point problems such as (9) *significantly differ* from GANs, see (Gazdieva et al., 2022, §6.2).

**Dataset transfer and domain adaptation**. Deep distance-based algorithms (Gretton et al., 2012; Long et al., 2015; 2017) or adversarial algorithms (Ganin & Lempitsky, 2015; Long et al., 2018) are common solutions to the domain adaptation problem. Using neural networks, these methods align probability distributions while maintaining the discriminativity between classes (Wang & Deng, 2018). Mostly they perform domain adaptation for image data at the feature level and are *typically not used at the pixel level* (data space). Pixel-level adaptation is typically performed by common unsupervised image-to-image *translation* techniques such as CycleGAN (Zhu et al., 2017; Hoffman et al., 2018; Almahairi et al., 2018) and UNIT (Huang et al., 2018; Liu et al., 2017).

**OT and domain adaptation.** Discrete OT solvers (EMD (Nash, 2000; Courty et al., 2016), Sinkhorn (Cuturi, 2013a), etc.) are usually employed to map labeled input samples to the unlabeled or partially labeled target samples. Combinations with neural feature extractors (Courty et al.,

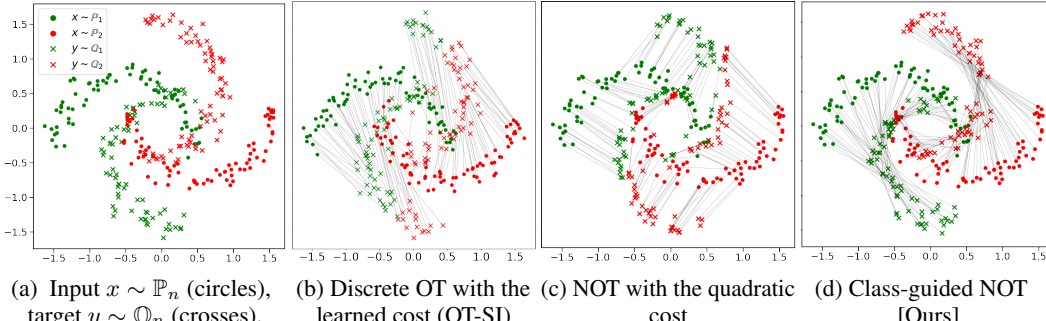

(a) Input $x \sim \mathbb{P}_n$ (circles), target $y \sim \mathbb{Q}_n$ (crosses).

(b) Discrete OT with the learned cost (OT-SI)

(c) NOT with the quadratic cost

(d) Class-guided NOT [Ours]

Figure 2: The results of mapping two moons using OT with different cost functionals.

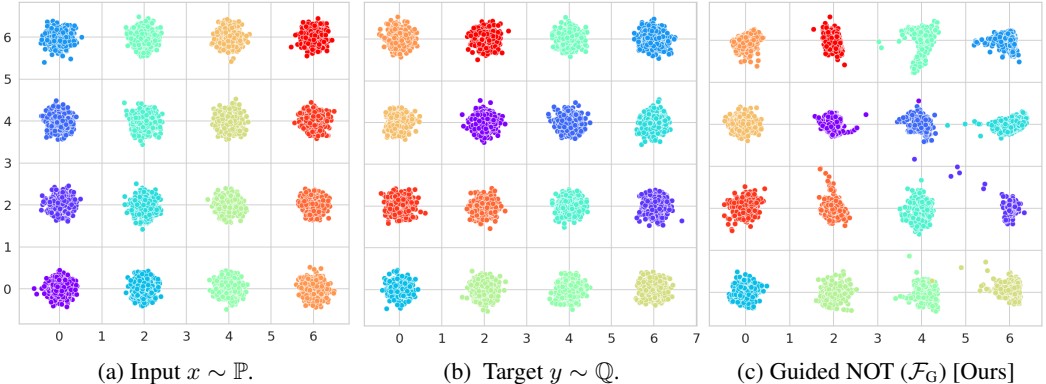

(a) Input $x \sim \mathbb{P}$.

(b) Target $y \sim \mathbb{Q}$.

(c) Guided NOT ($\mathcal{F}_G$) [Ours]

Figure 3: Illustration of the mapping between two Gaussian mixtures learned by our Algorithm 2.

2017; Redko et al., 2018; Bhushan Damodaran et al., 2018) are popular in domain adaptation (Courty et al., 2016). The available labels can be used to reconstruct the cost function and catch the data structure (Courty et al., 2016; Stuart & Wolfram, 2020; Liu et al., 2020; Li et al., 2019). Discrete OT performs a _matching_ between the given empirical samples and does not provide out-of-sample estimates. In contrast, our method generalizes to unseen source data and _generates_ new target data.

**Conditional generative models.** Conditional models, e.g., GAN (Mirza & Osindero, 2014), Adversarial Autoencoder (Makhzani et al., 2015), use the labels to apply conditional generation. They are _not relevant_ to our work as we do not use any label information during the inference. Our learned mapping is based only on the input content.

## 6 EXPERIMENTS

In this section, we test NOT with our cost functional $\mathcal{F}_G$ on toy cases (§6.1) and image data (§6.2). For comparison, we consider NOT with Euclidean costs (Korotin et al., 2022c; Rout et al., 2022; Korotin et al., 2021b; Fan et al., 2021a) and image-to-image translation models (Isola et al., 2017; Huang et al., 2018; Liu et al., 2017). The code is written in _PyTorch_ framework and will be made public along with the trained networks. On the image data, our method converges in 5-15 hours on a Tesla V100 (16 GB). We give the training details (architectures, pre-processing, etc.) in Appendix B. Our Algorithm 2 learns stochastic (_one-to-many_) transport maps $T(x, z)$. Following (Korotin et al., 2021b, §5), we also test deterministic $T(x, z) \equiv T(x)$, i.e., do not add a random noise $z$ to input. This disables stochasticity and yields deterministic (_one-to-one_) transport maps $x \mapsto T(x)$. In §6.1, (toy examples), we test only deterministic variant of our method. In §6.2, we test both cases.

### 6.1 TOY EXAMPLES

**The moons.** The task is to map two balanced classes of moons (red and green) between $\mathbb{P}$ and $\mathbb{Q}$ (circles and crosses in Figure 2a, respectively). The target distribution $\mathbb{Q}$ is $\mathbb{P}$ rotated by 90 degrees. The number of randomly picked labeled samples in each target moon **10**. The maps with learned by NOT with the quadratic cost ($\mathbb{W}_2$, (Korotin et al., 2022c; Rout et al., 2022; Fan et al., 2021a)) and our functional $\mathcal{F}_G$ are given in Figures 2c and 2d, respectively. For completeness, in Figure 2b we show the _matching_ performed by a _discrete_ OT-SI algorithm which learns the transport cost with a neural

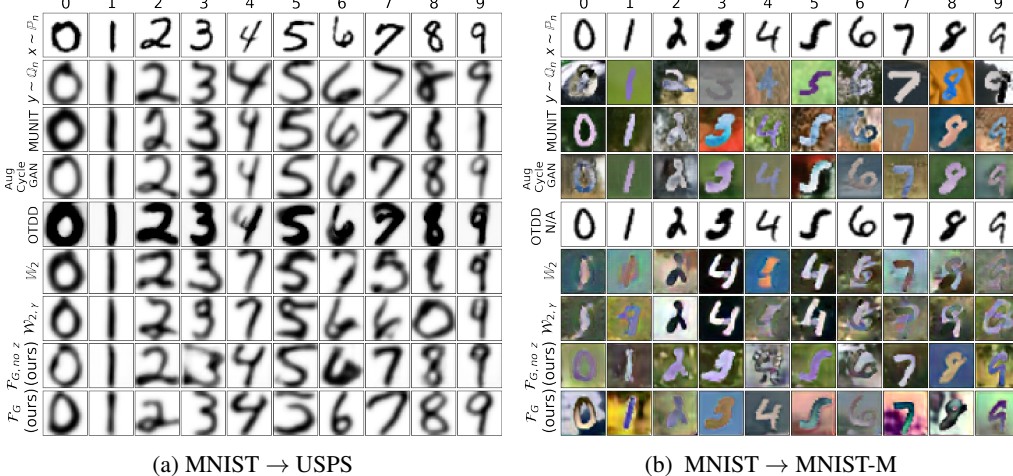

(a) MNIST → USPS        (b) MNIST → MNIST-M

Figure 4: The results of mapping between two similar images datasets (**related domains**).

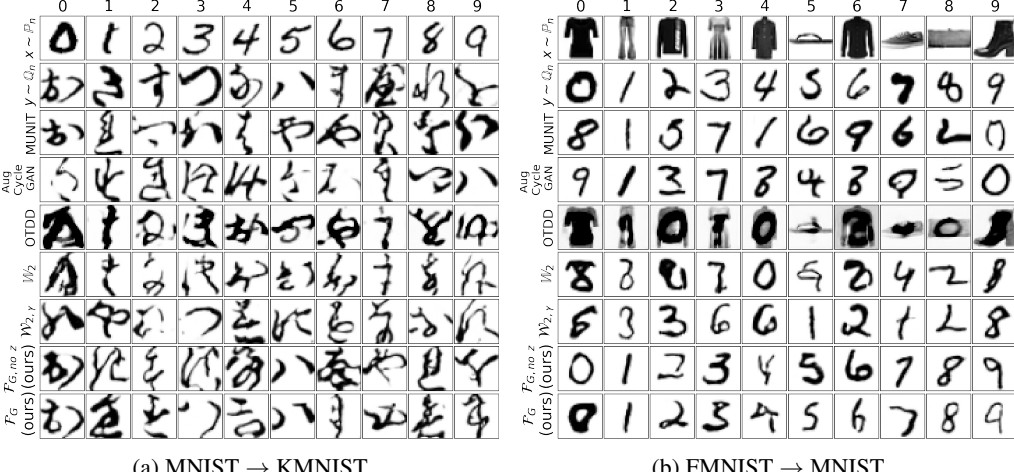

(a) MNIST → KMNIST        (b) FMNIST → MNIST

Figure 5: The results of mapping between two notably different datasets (**unrelated domains**).

net from a known classes' correspondence (Liu et al., 2020). The map for $\mathbb{W}_2$ does not preserve the classes (Figure 2c), while our map solves the task (Figure 2d).

**The Gaussians.** Here both $\mathbb{P}, \mathbb{Q}$ are balanced mixtures of 16 Gaussians, and each color denotes a unique class. The goal is to map Gaussians in $\mathbb{P}$ (Figure 3a) to respective Gaussians in $\mathbb{Q}$ (Figure 3b) which have the same color. The result of our method (**10** known target labels per class) is given in Figure 3c. It correctly maps the classes. NOT for the quadratic cost is not shown as it results in the *identity map* (the same image as Figure 3a) which is completely *mistaken* in classes.

## 6.2 IMAGE DATA EXPERIMENTS

**Datasets.** We use MNIST (LeCun & Cortes, 2010), MNIST-M (Ganin & Lempitsky, 2015), Fashion-MNIST (Xiao et al., 2017), KMNIST (Clanuwat et al., 2018) datasets as $\mathbb{P}, \mathbb{Q}$. Each dataset has 10 (balanced) classes and the pre-defined train-test split. We consider two cases. In the related domains case, source and target are close: MNIST → USPS, MNIST → MNIST-M. In the unrelated domains case, they notably differ: MNIST → KMNIST and FashionMNIST → MNIST. In all the cases, we use the default class correspondence between the datasets. For completeness, we provide an example with *imbalanced classes* and a *non-default* correspondence in Appendices E and G, respectively.

**Baselines:** We compare our method to principal unsupervised translation models. We consider (one-to-many) AugCycleGAN(Almahairi et al., 2018), MUNIT(Huang et al., 2018). We use the official implementations with the hyperparameters from the respective papers. Also we test NOT with Euclidean cost functions: the quadratic cost $\frac{1}{2}\|x - y\|^2$ ($\mathbb{W}_2$) and the $\gamma$-weak (one-to-many) quadratic cost ($\mathcal{W}_{2,\gamma}, \gamma = \frac{1}{10}$). The above-mentioned methods are unsupervised, i.e., they do not use the label information. For completeness, we add (one-to-one) OTDD flow (Alvarez-Melis & Fusi,

| Datasets ($32 \times 32$) | Image-to-Image Translation | | Flows | Neural Optimal Transport | | | |
|---|---|---|---|---|---|---|---|
| | MUNIT | Aug CycleGAN | OTDD | $\mathbb{W}_2$ | $\mathcal{W}_{2,\gamma}$ | $\mathcal{F}_G$, no $z$ [Ours] | $\mathcal{F}_G$ [Ours] |
| MNIST → USPS | 82.97 | **96.52** | 82.50 | 56.27 | 34.33 | 93.42 | 95.14 |
| MNIST → MNIST-M | 97.95 | **98.2** | - | 38.77 | 37.0 | 95.27 | 94.62 |
| MNIST → KMNIST | 12.27 | 8.99 | 4.46 | 6.13 | 6.82 | **79.20** | 61.91 |
| FMNIST → MNIST | 8.93 | 12.03 | 10.28 | 10.96 | 8.02 | 82.79 | **83.22** |

Table 1: Accuracy↑ of the maps learned by the translation methods in view.

| Datasets ($32 \times 32$) | MUNIT | Aug CycleGAN | OTDD | $\mathbb{W}_2$ | $\mathcal{W}_{2,\gamma}$ | $\mathcal{F}_G$, no $z$ [Ours] | $\mathcal{F}_G$ [Ours] |
|---|---|---|---|---|---|---|---|
| MNIST → USPS | 6.86 | 22.74 | > 100 | 4.60 | 3.05 | 5.40 | **2.87** |
| MNIST → MNIST-M | 11.68 | 26.87 | - | 19.43 | 17.48 | 18.56 | **6.67** |
| MNIST → KMNIST | 8.81 | 62.19 | > 100 | 12.85 | **9.46** | 17.26 | 9.69 |
| FMNIST → MNIST | 7.91 | 26.35 | > 100 | 7.51 | 7.02 | 7.14 | **5.26** |

Table 2: FID↓ of the samples generated by the translation methods in view.

2021; 2020). The method employs gradient flows to perform the transfer preserving the class label. Additionally, we show the results of ICNN-based $\mathbb{W}_2$ OT method (Makkuva et al., 2019; Korotin et al., 2021a) in Appendix F.

**Metrics.** All the models are fitted on the train parts of datasets; all the provided qualitative and quantitative results are *exclusively* for test (unseen) data. To evaluate the **visual quality**, we compute FID (Heusel et al., 2017) of the mapped source test set w.r.t. the target test set. To estimate the **accuracy** of the mapping we pre-train ResNet18 (He et al., 2016) classifier on the target dataset. We consider the mapping $T$ correct, if the predicted label for the mapped sample $T(x, z)$ matches the corresponding label of $x$.

**Results.** Qualitative results are show in Figures 4, 5; FID, accuracy – in Tables 2 and 1, respectively. To keep the figures simple, for all the models (one-to-one, one-to-many), we plot a single output per an input. For completeness, we show *multiple outputs* per an input for our method in Appendix C. Note that in our method and OTDD, we use **10** labeled samples per each class in the target distribution. The other methods under consideration do not use the label information.

In the *related domains case* (Figure 4), GAN-based methods and NOT with our guided cost $\mathcal{F}_G$ show high accuracy $\geq 90\%$. However, NOT with strong and weak quadratic costs provides low accuracy (35-50%). We presume that this is because for these dataset pairs the ground truth OT map for the (pixel-wise) quadratic cost simply does not preserve the class. This agrees with (Daniels et al., 2021, Figure 3) which test an entropy-regularized quadratic cost in a similar MNIST→USPS setup. For our method with guided cost $\mathcal{F}_G$, ablation study on $Z$ size presented in Appendix D. The OTDD gradient flows method provides reasonable accuracy on MNIST→USPS. However, OTDD has much higher FID than the other methods. Visually, the OTDD results are comparable to (Alvarez-Melis & Fusi, 2021, Figure 3).

In the *unrelated domains case* (Figure 5), which is of our main interest, all the methods except our method do not preserve the class structure since they do not use label information. Consequently, their accuracy is around $10\%$ (random guess). OTDD does not preserve the class structure and generates samples with worse FID. Only NOT with our cost $\mathcal{F}_G$ preserves the class labels accurately.

**Conclusion.** Our approach provides FID scores which are better or comparable to principal image-to-image translation methods. In the related domain case (Figure 4), we provide comparable accuracy of class preservation. When the domains are unrelated, we notably outperform existing approaches.

## 6.3 DISCUSSION

**Potential Impact.** Our method is a generic tool to learn transport maps between data distributions. In contrast to many other generative models, it allows to *control* the properties of the learned map via choosing a task-specific cost functional $\mathcal{F}$. Our method could be used for data generation and editing purposes, and analogously to GANs, have promising positive real-world applications, such as digital content creation and artistic expression. At the same time, generative models can also be used for negative data manipulation purposes such as deepfakes. In general, the impact of our work on society depends on the scope of its application and the task at the hand.

**Limitations.** To apply our method, one has to provide an estimator $\widehat{\mathcal{F}}(T)$ for the functional $\mathcal{F}$ which may be non-trivial. Besides, the construction of a cost functional $\mathcal{F}$ for a particular downstream task may be not straightfoward. This should be taken into account when using the method in practice. Constructing task-specific functionals $\mathcal{F}$ and estimators $\widehat{\mathcal{F}}$ is a promising future research avenue.

## 7 REPRODUCIBILITY

To ensure the reproducibility of our experiments, we provide the source code in the supplementary material. For toy experiments §6.1, run `twomoons_toy.ipynb` and `gaussian_toy.ipynb`. For the dataset transfer experiments §6.2, run `dataset_transfer.ipynb` and `dataset_transfer_no_z.ipynb`. The detailed information about the data preprocessing and training hyperparameters is presented in §6 and Appendix B.

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

## A    Proofs

*Proof of Theorem 1.* We use the dual form (6) to derive

$$\text{Cost}(\mathbb{P}, \mathbb{Q}) = \sup_v \left\{ \sup_u \left[ \int_{\mathcal{X}} u(x) d\mathbb{P}(x) - \mathcal{F}^*(u \oplus v) \right] + \int_{\mathcal{Y}} v(y) d\mathbb{Q}(y) \right\} = \quad (13)$$

$$\sup_v \left\{ \sup_u \left[ \int_{\mathcal{X}} u(x) d\mathbb{P}(x) - \sup_\pi \left( \int_{\mathcal{X} \times \mathcal{Y}} (u \oplus v) d\pi(x, y) - \mathcal{F}(\pi) \right) \right] + \int_{\mathcal{Y}} v(y) d\mathbb{Q}(y) \right\} = \quad (14)$$

$$\sup_v \left\{ \sup_u \left[ \int_{\mathcal{X}} u(x) d\mathbb{P}(x) + \inf_\pi \left( \mathcal{F}(\pi) - \int_{\mathcal{X} \times \mathcal{Y}} (u \oplus v) d\pi(x, y) \right) \right] + \int_{\mathcal{Y}} v(y) d\mathbb{Q}(y) \right\} = \quad (15)$$

$$\sup_v \left\{ \sup_u \inf_\pi \left( \mathcal{F}(\pi) - \int_{\mathcal{X}} u(x) d(\pi - \mathbb{P})(x) - \int_{\mathcal{Y}} v(y) d\pi(y) \right) + \int_{\mathcal{Y}} v(y) d\mathbb{Q}(y) \right\} \leq \quad (16)$$

$$\sup_v \left\{ \sup_u \inf_{\pi \in \Pi(\mathbb{P})} \left( \mathcal{F}(\pi) - \int_{\mathcal{X}} u(x) d(\pi - \mathbb{P})(x) - \int_{\mathcal{Y}} v(y) d\pi(y) \right) + \int_{\mathcal{Y}} v(y) d\mathbb{Q}(y) \right\} = \quad (17)$$

$$\sup_v \left\{ \sup_u \inf_{\pi \in \Pi(\mathbb{P})} \left( \mathcal{F}(\pi) - \int_{\mathcal{Y}} v(y) d\pi(y) \right) + \int_{\mathcal{Y}} v(y) d\mathbb{Q}(y) \right\} = \quad (18)$$

$$\sup_v \left\{ \inf_{\pi \in \Pi(\mathbb{P})} \left( \mathcal{F}(\pi) - \int_{\mathcal{Y}} v(y) d\pi(y) \right) + \int_{\mathcal{Y}} v(y) d\mathbb{Q}(y) \right\} \leq \quad (19)$$

$$\sup_v \left\{ \mathcal{F}(\pi^*) - \int_{\mathcal{Y}} v(y) \underbrace{d\pi^*(y)}_{d\mathbb{Q}(y)} + \int_{\mathcal{Y}} v(y) d\mathbb{Q}(y) \right\} = \mathcal{F}(\pi^*) = \text{Cost}(\mathbb{P}, \mathbb{Q}). \quad (20)$$

In line (13), we group the terms involving the potential $u$. In line (14), we express the conjugate functional $\mathcal{F}^*$ by using its definition. In transition to line (15), we replace $\inf_\pi$ operator with the equivalent $\sup_\pi$ operator with the changed sign. In transition to (16), we put the term $\int_{\mathcal{X}} u(x) d\mathbb{P}(x)$ under the $\inf_\pi$ operator; we use definition $(u \oplus v)(x, y) = u(x) + v(y)$ to split the integral over $\pi(x, y)$ into two separate integrals over $\pi(x)$ and $\pi(y)$ respectively. In transition to (17), we restrict the inner $\inf_\pi$ to probability measures $\pi \in \Pi(\mathbb{P})$ which have $\mathbb{P}$ as the first marginal, i.e. $d\pi(x) = d\mathbb{P}(x)$. This provides an upper bound on (16), in particular, all $u$-dependent terms vanish, see (18). As a result, we remove the $\sup_u$ operator in line (19). In transition to line (20) we substitute an optimal plan $\pi^* \in \Pi(\mathbb{P}, \mathbb{Q}) \subset \Pi(\mathbb{Q})$ to upper bound (19). Since $\text{Cost}(\mathbb{P}, \mathbb{Q})$ turns to be both an upper bound (20) and a lower bound (13) for (19), we conclude that (7) holds. □

*Proof of Theorem 2.* Assume that $\pi^* \notin \arg\inf_{\pi \in \Pi(\mathbb{P})} \mathcal{L}(v^*, \pi)$, i.e.,

$$\mathcal{L}(v^*, \pi^*) > \inf_{\pi \in \Pi(\mathbb{P})} \mathcal{L}(v^*, \pi) = \text{Cost}(\mathbb{P}, \mathbb{Q}).$$

We substitute $v^*$ and $\pi^*$ to $\mathcal{L}$ and see that

$$\mathcal{L}(v^*, \pi^*) = \left[ \mathcal{F}(\pi^*) - \int_{\mathcal{Y}} v(y) \underbrace{d\pi^*(y)}_{d\mathbb{Q}(y)} \right] + \int_{\mathcal{Y}} v(y) d\mathbb{Q}(y) = \mathcal{F}(\pi^*) = \text{Cost}(\mathbb{P}, \mathbb{Q}),$$

which is a contradiction. Thus, the assumption is wrong and (8) holds. □

*Proof of Proposition 3.* First, we prove that it is *-separately increasing. For $\pi \in \mathcal{M}(\mathcal{X} \times \mathcal{Y}) \setminus \Pi(\mathbb{P})$ it holds that $\mathcal{F}(\pi) = +\infty$. Consequently,

$$\int_{\mathcal{X} \times \mathcal{Y}} c(x, y) d\pi(x, y) - \mathcal{F}(\pi) = \int_{\mathcal{X} \times \mathcal{Y}} (u(x) + v(y)) d\pi(x, y) - \mathcal{F}(\pi) = -\infty. \quad (21)$$

When $\pi \in \Pi(\mathbb{P})$ it holds that $\pi$ is a probability measure. We integrate $u(x) + v(y) \leq c(x, y)$ w.r.t. $\pi$, substract $\mathcal{F}(\pi)$ and obtain

$$\int_{\mathcal{X} \times \mathcal{Y}} c(x, y) d\pi(x, y) - \mathcal{F}(\pi) \geq \int_{\mathcal{X} \times \mathcal{Y}} (u(x) + v(y)) d\pi(x, y) - \mathcal{F}(\pi). \quad (22)$$

By taking the $\sup$ of (21) and (22) w.r.t. $\pi \in \mathcal{M}(\mathcal{X} \times \mathcal{Y})$, we obtain $\mathcal{F}^*(c) \geq \mathcal{F}^*(u \oplus v)$.[2]

Next, we prove that $\mathcal{F}$ is convex. We prove that every term $\mathcal{E}^2\big(T_\pi \sharp(\mathbb{P}_n \times \mathbb{S}), \mathbb{Q}_n\big)$ is convex in $\pi$.

**First**, we show that $\pi \mapsto f_n(\pi) \stackrel{def}{=} T_\pi \sharp(\mathbb{P}_n \times \mathbb{S})$ is *linear* in $\pi \in \Pi(\mathbb{P})$.

Pick any $\pi_1, \pi_2, \pi_3 \in \Pi(\mathbb{P})$ which lie on the same line. Without loss of generality we assume that $\pi_3 \in [\pi_1, \pi_2]$, i.e., $\pi_3 = \alpha \pi_1 + (1-\alpha)\pi_2$ for some $\alpha \in [0, 1]$. We need to show that

$$f_n(\pi_3) = \alpha f_n(\pi_1) + (1-\alpha)f_n(\pi_2). \tag{23}$$

In what follows, for a random variable $U$ we denote its distribution by $\mathrm{Law}(U)$.

The first marginal distribution of each $\pi_i$ is $\mathbb{P}$. From the gluing lemma (Villani, 2008, §1) it follows that there exists a triplet of (dependent) random variables $(X, Y_1, Y_2)$ such that $\mathrm{Law}(X, Y_i) = \pi_i$ for $i = 1, 2$. We define $Y_3 = Y_r$, where $r$ is an *independent* random variable which takes values in $\{1, 2\}$ with probabilities $\{\alpha, 1 - \alpha\}$. From the construction of $Y_3$ it follows that $\mathrm{Law}(X, Y_3)$ is a mixture of $\mathrm{Law}(X, Y_1) = \pi_1$ and $\mathrm{Law}(X, Y_2) = \pi_2$ with weights $\alpha$ and $1 - \alpha$. Thus, $\mathrm{Law}(X, Y_3) = \alpha \pi_1 + (1-\alpha)\pi_2 = \pi_3$. We conclude that $\mathrm{Law}(Y_3 | X = x) = \pi_3(\cdot | x)$ for $\mathbb{P}$-almost all $x \in \mathcal{X}$ (recall that $\mathrm{Law}(X) = \mathbb{P}$). On the other hand, again by the construction, the conditional $\mathrm{Law}(Y_3 | X = x)$ is a mixture of $\mathrm{Law}(Y_1 | X = x) = \pi_1(\cdot | x)$ and $\mathrm{Law}(Y_2 | X = x) = \pi_2(\cdot | x)$ with weights $\alpha$ and $1 - \alpha$. Thus, $\pi_3(\cdot | x) = \alpha \pi_1(\cdot | x) + (1-\alpha)\pi_2(\cdot | x)$ holds true for $\mathbb{P}$-almost all $x \in \mathcal{X}$.

Consider independent random variables $X_n \sim \mathbb{P}_n$ and $Z \sim \mathbb{S}$. From the definition of $T_{\pi_i}$ we conclude that $\mathrm{Law}\big(T_{\pi_i}(x, Z)\big) = \pi_i(\cdot | x)$ for $\mathbb{P}$-almost all $x \in \mathcal{X}$ and, since $\mathbb{P}_n$ is a component of $\mathbb{P}$, for $\mathbb{P}_n$-almost all $x \in \mathcal{X}$ as well. As a result, we define $T_i = T_{\pi_i}(X_n, Z)$ and derive

$$\mathrm{Law}(T_3 | X_n = x) = \pi_3(\cdot | x) = \alpha \pi_1(\cdot | x) + (1 - \alpha)\pi_2(\cdot | x) =$$
$$\alpha \mathrm{Law}(T_1 | X_n = x) + (1 - \alpha)\mathrm{Law}(T_2 | X_n = x)$$

for $\mathbb{P}_n$-almost all $x \in \mathcal{X}$. Thus, $\mathrm{Law}(X_n, T_3)$ is also a mixture of $\mathrm{Law}(X_n, T_1)$ and $\mathrm{Law}(X_n, T_2)$ with weights $\alpha$ and $1 - \alpha$. In particular, $\mathrm{Law}(T_3) = \alpha \mathrm{Law}(T_1) + (1 - \alpha)\mathrm{Law}(T_2)$. We note that $\mathrm{Law}(T_i) = f_n(\pi_i)$ by the definition of $f_n$ and obtain (23).

**Second**, we highlight that for every $\nu \in \mathcal{P}(\mathcal{Y})$, the functional $\mathcal{P}(\mathcal{Y}) \ni \mu \to \mathcal{E}^2(\mu, \nu)$ is convex in $\mu$. Indeed, $\mathcal{E}^2$ is a particular case of (the square of) Maximum Mean Discrepancy (MMD, (Sejdinovic et al., 2013)). Therefore, there exists a Hilbert space $\mathcal{H}$ and a function $\phi : \mathcal{Y} \to \mathcal{H}$ (feature map), such that

$$\mathcal{E}^2(\mu, \nu) = \left\| \int_{\mathcal{Y}} \phi(y) d\mu(y) - \int_{\mathcal{Y}} \phi(y) d\nu(y) \right\|_{\mathcal{H}}^2.$$

Since the kernel mean embedding $\mu \mapsto \int_{\mathcal{Y}} \phi(y) d\mu(y)$ is linear in $\mu$ and $\|\cdot\|_{\mathcal{H}}^2$ is convex, we conclude that $\mathcal{E}^2(\mu, \nu)$ is convex in $\mu$. To finish the proof, it remains to combine the fact that $\pi \mapsto T_\pi \sharp(\mathbb{P}_n \times \mathbb{S})$ is linear and $\mathcal{E}^2(\cdot, \mathbb{Q}_n)$ is convex in the first argument. $\qquad\square$

*Proof of Proposition 1.* Direct calculation of the expectation of (12) yields the value

$$\mathbb{E}\|Y - T(X, Z)\| - \frac{1}{2}\mathbb{E}\|T(X, Z) - T(X', Z')\| =$$

$$\mathbb{E}\|Y - T(X, Z)\| - \frac{1}{2}\mathbb{E}\|T(X, Z) - T(X', Z')\| - \frac{1}{2}\mathbb{E}\|Y - Y'\| + \frac{1}{2}\mathbb{E}\|Y - Y'\| =$$

$$\mathcal{E}^2\big(T\sharp(\mathbb{P}_n \times \mathbb{S}), \mathbb{Q}_n\big) + \frac{1}{2}\mathbb{E}\|Y - Y'\|, \tag{24}$$

where $Y, Y' \sim \mathbb{Q}_n$ and $(X, Z), (X', Z') \sim (\mathbb{P}_n \times \mathbb{S})$ are independent random variables. It remains to note that $\frac{1}{2}\mathbb{E}\|Y - Y'\|$ is a $T$-independent constant. $\qquad\square$

---

[2]The proof is generic and works for any functional which equals $+\infty$ outside $\pi \in \mathcal{P}(\mathcal{X} \times \mathcal{Y})$.

# B EXPERIMENTS DETAILS

DATA PREPOSSESSING. We rescale the images to size 32×32 and normalize their channels to $[-1, 1]$. For the grayscale images, we repeat their channel 3 times and work with 3-channel images. We do not apply any augmentations to data. We use the default train-test splits for all the datasets.

TRAINING DETAILS. In our Algorithm 2, we use Adam (Kingma & Ba, 2014) optimizer with $lr = 10^{-4}$ for both $T_\theta$ and $v_\omega$. The number of inner iterations for $T_\theta$ is $K_T = 10$. Doing preliminary experiments, we noted that it is sufficient to use small mini-batch sizes $K_X, K_Y, K_Z$ in (12). Therefore, we decided to average loss values over $K_B$ small independent mini-batches (each from class $n$ with probability $\alpha_n$) rather than use a single large batch from one class. This is done parallel with tensor operations.

**Two moons**. We use 500 train and 150 test samples for each moon. We use the fully-connected net with 2 ReLU hidden layers size of 128 for both $T_\theta$ and $v_\omega$. We train the model for 10k iterations of $v_\omega$ with $K_B = 32, K_X = K_Y = 2$ ($K_Z$ plays no role as we do not use $z$ here).

**Gaussians.** We use the fully-connected network with 2 ReLU hidden layers size of 256 for both $T_\theta$ and $v_\omega$. There are 10000 train and 500 test samples in each Gaussian. We train the model for 10k iterations of $v_\omega$ with $K_B = 32, K_X = K_Y = 2$ ($K_Z$ plays no role here as well).

**Images.** We use WGAN-QC discriminator's ResNet architecture (He et al., 2016) for potential $v_\omega$. We use UNet[3] (Ronneberger et al., 2015) as the stochastic transport map $T_\theta(x, z)$. To condition it on $z$, we insert conditional instance normalization (CondIN) layers after each UNet's upscaling block[4]. We use CondIN from AugCycleGAN (Almahairi et al., 2018). In experiments, $z$ is the 128-dimensional standard Gaussian noise.

The batch size is $K_B = 32, K_X = K_Y = 2, K_Z = 2$ for training with $z$. When training without $z$, we use the original UNet without conditioning; the batch parameters are the same ($K_Z$ does not matter). Our method converges in $\approx$ 60k iterations of $v_\omega$.

For comparison in image domain we use the official implementations with the hyperparameters from the respective papers: AugCycleGAN[5] (Almahairi et al., 2018), MUNIT[6](Huang et al., 2018). For comparison with NOT ($\mathbb{W}_2, \mathcal{W}_{2,\gamma}$), we use the code shared by the authors of (Korotin et al., 2022c).

**OTDD flow details.** As in our method, the number of labeled samples in each class is 10. We learn the OTDD flow between the labeled source dataset[7] and labeled target samples. Note the OTDD method does not use the unlabeled target samples. As the OTDD method does not produce out-of-sample estimates, we train UNet to map the source data to the data produced by the OTDD flow via regression. Then we compute the metrics on test (FID, accuracy) for this mapping network.

---

[3]`github.com/milesial/Pytorch-UNet`
[4]`github.com/kgkgzrtk/cUNet-Pytorch`
[5]`github.com/aalmah/augmented_cyclegan`
[6]`github.com/NVlabs/MUNIT`
[7]We use only 15k source samples since OTDD is computationally heavy (the authors use 2k samples).

# C ADDITIONAL EXAMPLES OF STOCHASTIC MAPS

In this subsection, we provide additional examples the learned stochastic map for $\mathcal{F}_G$ (with $z$).

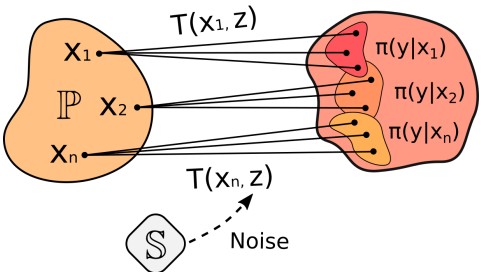

Figure 6: Implicit representation of $\pi \in \Pi(\mathbb{P})$ via function $T = T_\pi : \mathcal{X} \times \mathcal{Z} \to \mathcal{Y}$.

We consider all the image datasets from the main experiments (§6). The results are shown in Figure 7 and demonstrate that for a fixed $x$ and different $z$, our model generates diverse samples.

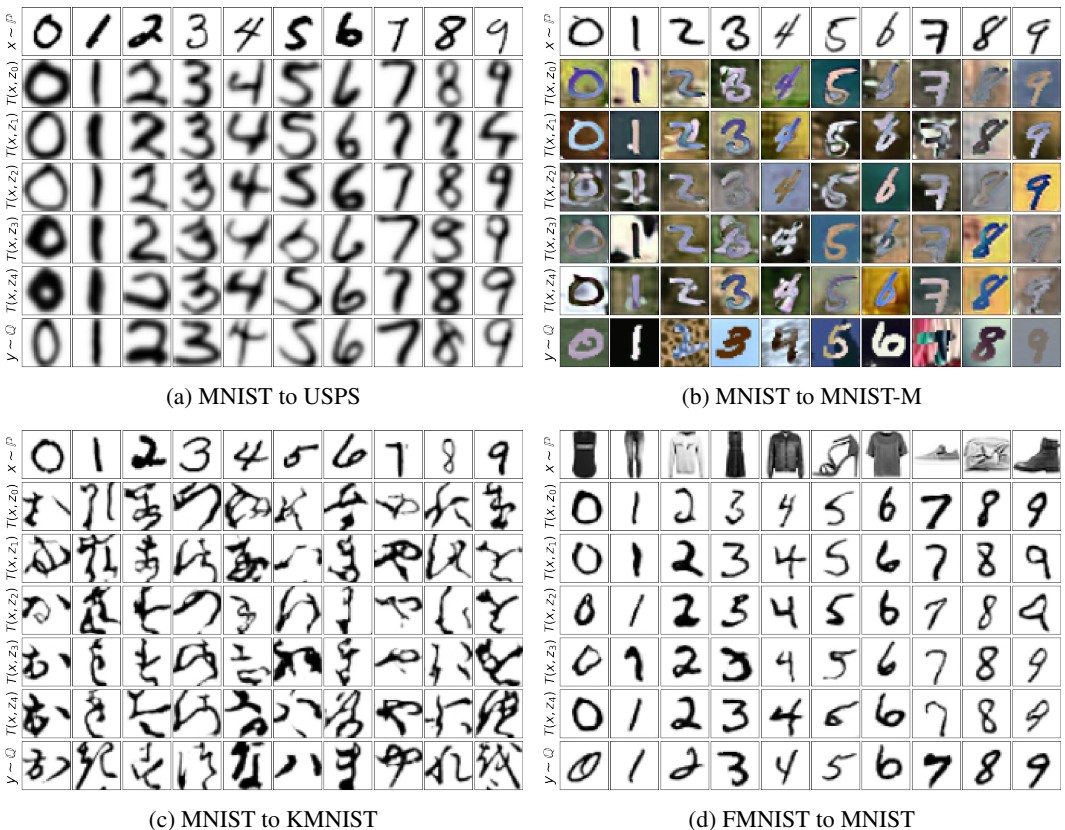

(a) MNIST to USPS

(b) MNIST to MNIST-M

(c) MNIST to KMNIST

(d) FMNIST to MNIST

Figure 7: Stochastic transport maps $T_\theta(x, z)$ learned by our Algorithm 2. Additional examples.

# D ABLATION STUDY OF THE LATENT SPACE DIMENSION

In this subsection, we study the structure of the learned stochastic map for $\mathcal{F}_G$ with different latent space dimensions $Z$. We consider MNIST $\rightarrow$ USPS transfer task (10 classes). The results are shown in Figures 8, 9 and Table 3. As it can be seen, our model performs comparably for different $Z$.

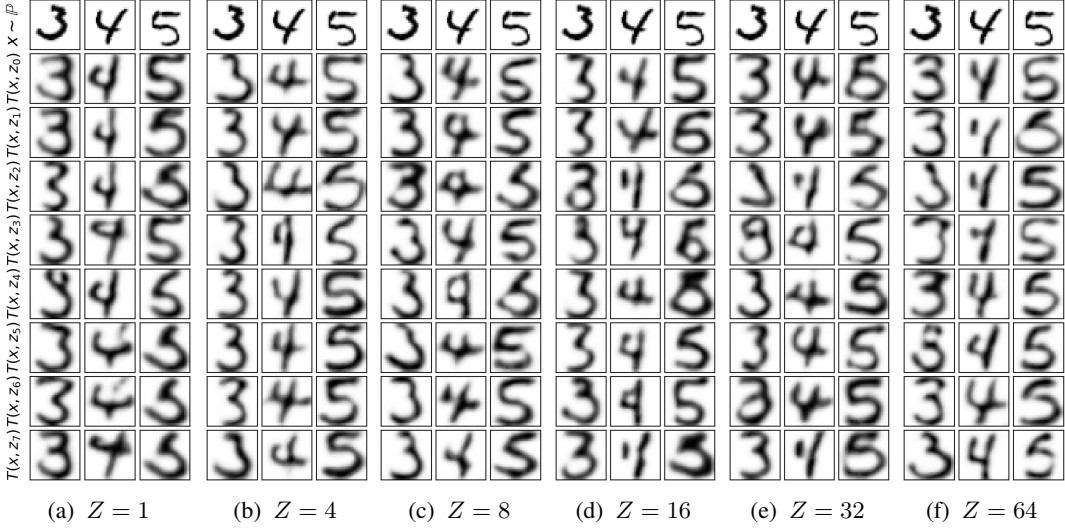

| (a) $Z=1$ | (b) $Z=4$ | (c) $Z=8$ | (d) $Z=16$ | (e) $Z=32$ | (f) $Z=64$ |

Figure 8: MNIST $\rightarrow$ USPS translation with functional $\mathcal{F}_G$ and varying $Z = 1, 4, 8, 16, 32, 64$.

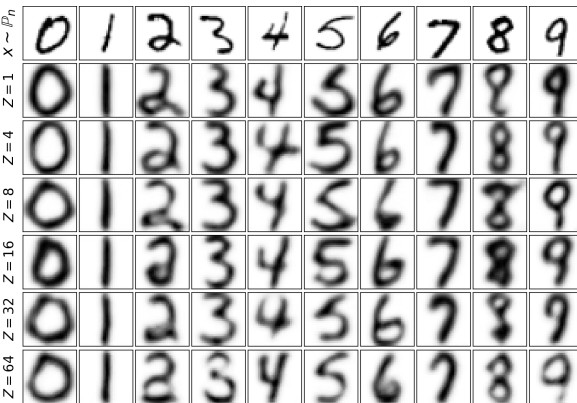

Figure 9: Stochastic transport maps $T_\theta(x, z)$ learned by our Algorithm 2 with different sizes of $Z$.

| Metrics | $Z=1$ | $Z=4$ | $Z=8$ | $Z=16$ | $Z=32$ | $Z=64$ |
|---|---|---|---|---|---|---|
| Accuracy | 86.96 | 93.48 | 91.82 | 92.08 | 92.25 | 92.95 |
| FID | 4.90 | 5.88 | 4.63 | 3.80 | 4.34 | 4.61 |

Table 3: Accuracy↑ and FID↓ of the stochastic maps MNIST $\rightarrow$ USPS learned by the our translation method with different noise size $Z$.

# E IMBALANCED CLASSES

In this subsection, we study the behaviour of the optimal map for $\mathcal{F}_G$ when the classes are imbalanced in input and target domains. Since out method learns a transport map from $\mathbb{P}$ to $\mathbb{Q}$, it should capture the class balance of the $\mathbb{Q}$ *regardless* of the class balance in $\mathbb{P}$. We check this below.

We consider MNIST $\to$ USPS datasets with $n = 3$ classes in MNIST and $n = 3$ classes in USPS. We assume that the class probabilities are $\alpha_1 = \alpha_2 = \frac{1}{2}$, $\alpha_3 = 0$ and $\beta_1 = \beta_2 = \beta_3 = \frac{1}{3}$. That is, there is no class 3 in the source dataset and it is not used anywhere during training. In turn, the target class 3 is not used when training $T_\theta$ but is used when training $f_\omega$. All the hyperparameters here are the same as in the previous MNIST $\to$ USPS experiments with 10 known labels in target classes. The results are shown in Figure 10a and 11a. We present deterministic (no $z$) and stochastic (with $z$) maps.

Our cost functional $\mathcal{F}_G$ stimulates the map to maximally preserve the input class. However, to transport $\mathbb{P}$ to $\mathbb{Q}$, the model *must* change the class balance. We show the confusion matrix for learned maps $T_\theta$ in Figures 10b, 11b. It illustrates that model maximally preserves the input classes $0, 1$ and uniformly distributes the input classes 0 and 1 into the class 2, as suggested by our cost functional.

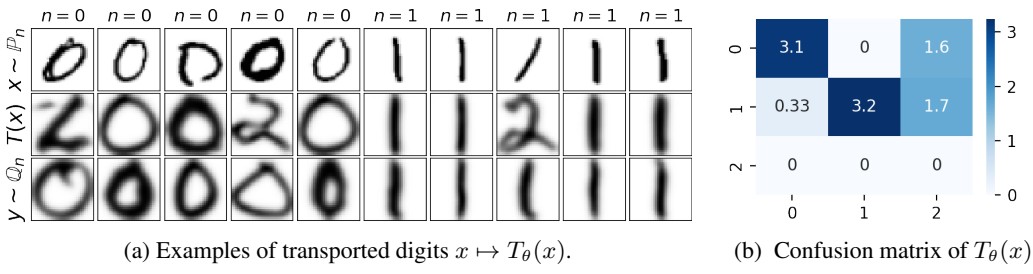

(a) Examples of transported digits $x \mapsto T_\theta(x)$.

(b) Confusion matrix of $T_\theta(x)$.

Figure 10: Imbalanced MNIST $\to$ USPS translation with functional $\mathcal{F}_G$ (deterministic, no $z$).

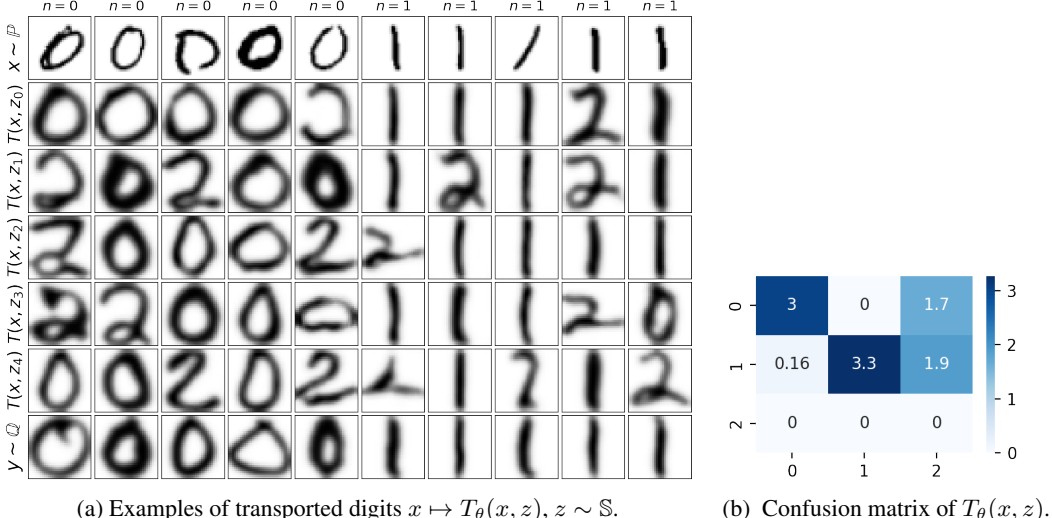

(a) Examples of transported digits $x \mapsto T_\theta(x, z)$, $z \sim \mathbb{S}$.

(b) Confusion matrix of $T_\theta(x, z)$.

Figure 11: Imbalanced MNIST $\to$ USPS translation with functional $\mathcal{F}_G$ (stochastic, with $z$).

## F ICNN-BASED DATASET TRANSFER

For completeness, we show the performance of ICNN-based method for the strong (2) quadratic transport cost $c(x, y) = \frac{1}{2}\|x - y\|^2$ on the dataset transfer task. We use the non-minimax version (Korotin et al., 2021a) of the ICNN-based method by (Makkuva et al., 2019). We employ the publicly available code and dense ICNN architectures from the Wasserstein-2 benchmark repository [8]. The batch size is $K_B = 32$, total number of iterations is 100k, $lr = 3 \cdot 10^{-3}$, and the Adam optimizer is used. The datasets are preprocessed as in the other experiments, see Appendix B.

The qualitative results for MNIST→USPS and FashionMNIST→MNIST transfer are given in Figure 12. The results are reasonable in the first case (related domains). However, they are visually unpleasant in the second case (unrelated domains). This is expected as the second case is notably harder. More generally, as derived in the Wasserstein-2 benchmark (Korotin et al., 2021b), the ICNN models do not work well in the pixel space due to the poor expressiveness of ICNN architectures. The ICNN method achieved **18.8**% accuracy and ≫**100** FID in the FMNIST→MNIST transfer, and **35.6**% and accuracy and **13.9** FID in the MNIST→USPS case. All the metrics are much worse than those achieved by our general OT method with the class-guided functional $\mathcal{F}_G$, see Table 1, 2 for comparison.

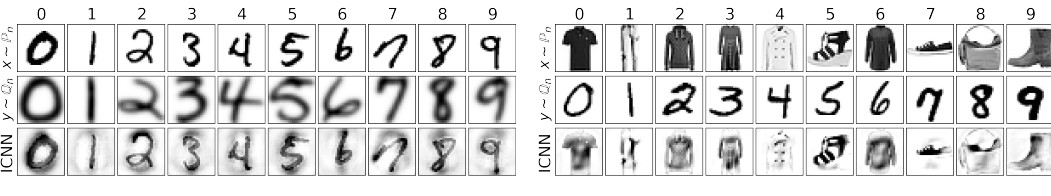

(a) MNIST→USPS transfer.          (b) FMNIST→MNIST transfer.

Figure 12: Results of ICNN-based method applied to the dataset transfer task.

## G NON-DEFAULT CLASS CORRESPONDENCE

To show that our method can work with any arbitrary correspondence between datasets, we also consider FMNIST→MNIST dataset transfer with the following non-default correspondence between the dataset classes:

$$0 \to 9, 1 \to 0, 2 \to 1, 3 \to 2, 4 \to 3, 5 \to 4, 6 \to 5, 7 \to 6, 8 \to 7, 9 \to 8.$$

In this experiment, we use the same architectures and data preprocessing as in dataset transfer tasks; see Appendix B. We use our $\mathcal{F}_G$ (11) as the cost functional and learn a deterministic transport map $T$ (no $z$). In this setting, our method produce comparable results to the previously reported in Section 6 accuracy equal to **83.1**, and FID **6.69**. The qualitative results are given in Figure 13.

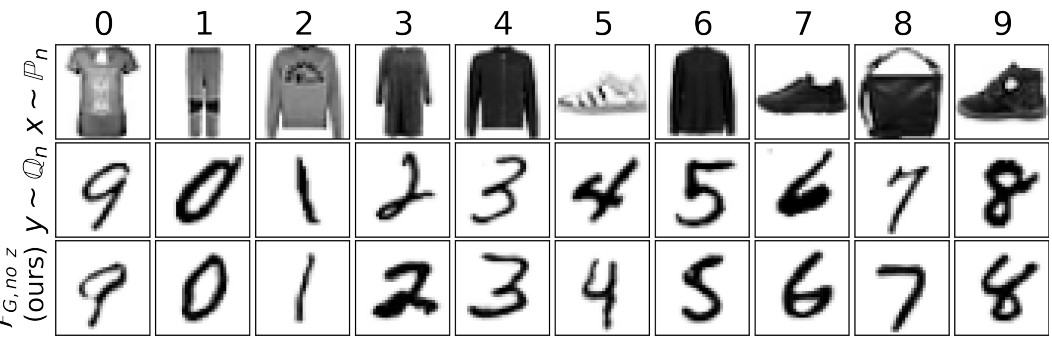

Figure 13: FMNIST→MNIST mapping with $\mathcal{F}_G$ no $z$ cost, classes are permuted.

---

[8]github.com/iamalexkorotin/Wasserstein2Benchmark

