# OpenReview forum: "Neural Optimal Transport with General Cost Functionals"
_ICLR.cc/2023/Conference — Submitted to ICLR 2023_

### Official Review · Reviewer_AhFA · 2022-10-22

**Confidence:** 4
**Correctness:** 3
**Technical Novelty And Significance:** 2
**Empirical Novelty And Significance:** 2
**Recommendation:** 6

**Clarity, Quality, Novelty And Reproducibility:**

This paper overall clearly demonstrated an approach to estimate optimal transport with neural networks. I would expect the authors to further elaborate on the contribution and distinguish this work from previous ones since the weak-Cost formulation and algorithm 1 are very similar to [Korotion 2022c]. The clarity is mostly good.


**Details Of Ethics Concerns:**

No ethics concerns.

**Strength And Weaknesses:**

Pro:
-This manuscript is pretty well-written and easy to follow.
-The proposed method is well motivated as the authors comprehensively introduced different OT problem settings step by step.
-The experimental results are well-organized. The authors provided both qualitative and quantitative results to evaluate their proposed method.

Cons and questions:
-There might be some concerns regarding the novelty. The proposed objective, as well as algorithm 1 are similar to that in one of the cited works [Korotin 2022c]. Although in this paper, the authors computed a class-guided cost.
-In equation (11) and Proposition 1. It seems that the stochastic map is asked to map one mixture towards another. However, how to define the correspondence of classes between different domains? What will happen if the number of classes is different between two datasets? This may break down to the optimal transport between two mixture distributions.


**Summary Of The Paper:**

The authors proposed to estimate optimal transport maps that are parameterized as neural networks.
The proposed method is established based on reformulating the Kantorovich dual problem as maxmin problem which optimizes for a stochastic transport map and a potential jointly. Then, the authors also introduce a cost metric that encourages the transport map to preserve the class information, when mapping two datasets with conditional labels.
The authors evaluated their proposed method on a few datasets, including toy synthetic data and real-world image datasets. The proposed method is shown to be effective.

**Summary Of The Review:**

Overall this paper well-presented. Although the proposed method seems to be a little bit incremental since it is built upon existing formulation and algorithms. Since the presentation quality is high, I lean towards acceptance but I am still on the fence.

---

> ### Author Response · Authors · 2022-11-13
> **Answers to the Reviewer AhFA**
>
> Thank you for your comments and questions. We took your feedback onboard and uploaded a revised version of our paper. Please find the answers to your questions below.
>
> **Q1: Algorithm 1 are similar to that in one of the cited works [Korotin 2022c]. I would expect the authors to further elaborate on the contribution and distinguish this work from previous ones since the weak-Cost formulation and algorithm 1 are very similar to [Korotion 2022c]. The proposed method seems to be a little bit incremental since it is built upon existing formulation and algorithms.**
>
> The related prior work (Korotin et al. 2022c) only considers optimization of strong and weak cost functionals. The current work develops an optimization algorithm for general cost functionals, allowing previously unreleased problems to be approached with computational optimal transport. In particular, we derive the duality formula for the general OT problem, which enables stochastic optimization, as well as the computational algorithm to solve it in practice. All of the theorems, corollaries, and propositions in Sections 3 and 4 on which our algorithm is based are original.
>
> Following your suggestion, we extended the discussion of the relation with (Korotin et al. 2022c). Please, see Section 3.3.
>
> **Q2: How to define the correspondence of classes between different domains?**
>
> The correspondence between classes in different domains depends on the task at hand. Usually, the correspondence can be set naturally: for example, MNIST-USPS and MNIST-MMNIST contain the same classes.
>
> For the experimental purposes, we also considered FMNIST-MNIST and MNIST-KMNIST datasets which are unrelated and there is no natural correspondence between the classes. This was done intentionally to stress-test our method. We built the correspondence using the default dataset labeling but any other class correspondence will work.
>
>
> **Q3: What will happen if the number of classes is different between two datasets?**
>
> In Appendix E, we investigate how the optimal map for $\mathcal{F}_G$ behaves when the classes are imbalanced in the input and target domains.
>
> We consider MNIST $\rightarrow$ USPS datasets with $n=3$  classes in MNIST and $n=3$ classes in USPS but with the class probabilities  $\alpha_{1}=\alpha_{2}=\frac{1}{2}$, $\alpha_{3}=0$ and $\beta_{1}=\beta_{2}=\beta_{3}=\frac{1}{3}$, i.e., there is **actually no class "2" in MNIST**. Our cost functional $\mathcal{F}_G$ stimulates the map to preserve the input class as much as possible. However, to transport $\mathbb{P}$ to $\mathbb{Q}$, the transport must change the class balance. Figures 10b and 11b in Appendix E show the confusion matrix for learned OT maps. We see that the model tries to maximally preserve the existing input classes "0" and "1" and uniformly distributes the missing class "2" into all the other classes.
>
>
> **Concluding remarks.** Please respond to our post to let us know if the clarifications above suitably address your concerns about our work. We are happy to address any remaining points during the discussion phase; if the responses above are sufficient, we kindly ask that you consider raising your score.
>
> **References**
>
> - Alexander Korotin, Daniil Selikhanovych, and Evgeny Burnaev. Neural optimal transport. arXiv preprint arXiv:2201.12220, 2022c.

---

### Official Review · Reviewer_tFkj · 2022-10-24

**Confidence:** 5
**Correctness:** 2
**Technical Novelty And Significance:** 2
**Empirical Novelty And Significance:** 2
**Recommendation:** 3

**Clarity, Quality, Novelty And Reproducibility:**

The overall structure of this paper is clear and the quality is average.
In particular, the key of this paper is add class information to neural optimal transport, lack novelty.
The experiments can Reproducibility.

**Strength And Weaknesses:**

Strength: A functional is constructed to map data distributions with preserving the class-wise structure of data.

Weakness: the general cost functionals lacks designed and selected in detail.
Learned stochastic OT map via the algorithm that may be fake solution.
In Image-to-Image Translation task, Lack of comparison with other neural network-based algorithms for solving OT mappings, such as using Input Convex Neural Networks to estimate OT mapping.

**Summary Of The Paper:**

This present a novel neural-networks-based algorithm to compute optimal transport (OT) plans and maps for general cost functionals.
The algorithm generalizes prior OT methods for weak and strong cost functionals.
And it constructs a functional to map data distributions with preserving the class-wise structure of data.

**Summary Of The Review:**

Although the overall structure of this paper is clear, it lacks novelty. The experimental results do not show the superiority of the algorithm, and it is difficult to design a specific task cost function, lack of guidance.

---

> ### Author Response · Authors · 2022-11-13
> **Answers to the Reviewer tFkj**
>
> Thank you for spending time reviewing our paper and providing valuable feedback that will help us improve the manuscript. Please find below the answers to your questions.
>
> **Q1: The general cost functionals lacks designed and selected in detail [...] and it is difficult to design a specific task cost function, lack of guidance.**
>
> As we discuss in Section 6, the cost functional $\mathcal{F}$, should be selected based on the task at hand. In our paper, we constructed (Section 4) and evaluated (Section 6) the functional $\mathcal{F}_G$ for the class-guided dataset transfer task. As we already noted in the limitations (Section 6.3), we hope that our research stimulates further work on creating task-specific cost functionals for a variety of other tasks.
>
> *Does our response answer your question? If not, we kindly ask to further elaborate your question.*
>
> **Q2: Stochastic OT map may be fake solution.**
>
> According to our paper's Corollary 3, if the functional $\mathcal{F}$ is **strictly** convex in $\pi$, all the solutions are true OT maps. After Corollary 3, we state that in order to ensure that all the solutions are OT maps, one may consider adding strictly convex regularizers to $\mathcal{F}$ with a small weight, such as the conditional kernel variance (Korotin et al., 2022b).
>
> Although our proposed functional $\mathcal{F}_G$ is not strictly convex, we found that this is not a problem in practice for class-guided transfer on a variety of datasets.
>
> **Q3: Additional comparisons, e.g., with OT methods based on Input-convex Neural Networks (ICNNs).**
>
> The ICNN-based methods find the OT map **only** for the strong quadratic transport cost (Wasserstein-2). The recent analysis of neural OT methods (Korotin et al., 2021b; Figure 5a) demonstrated that in image data problems, ICNN-based solvers do not perform well. We mentioned this in the initial submission, see Section 5 (related work).
>
> Moreover, we refer the reviewer to the recent paper "Variational Wasserstein gradient flow" (Fan et al. 2020), which was co-authored by Amirhossein Taghvaei who is the author of the principal ICNN-based techniques for OT (Taghvaei et al., 2019; Makkuva et al., 2020). In this paper, it is directly shown that regular parameterization (non-ICNN) leads to notably improved performance in an OT-related task, see their Figures 3a and 4a.
>
> Due to this, we **compared** our method to the strong quadratic transport cost OT ($\mathbb{W}_2$) method which is based on a **regular parameterization** (non-ICNN). Please see dataset transfer results in Figures 4, 5, and Tables 1 and 2.
>
> **Q4: Lacks novelty.**
>
> Both Section 3 (derivation of the algorithm), Section 4 (class-guided functional $\mathcal{F}_G)$ and Section 6 (application for the class-guided transfer) which constitute our main contribution provide **novel** results: theorems, corollaries, propositions and experiments.
>
> **Q5: The experimental results do not show the superiority of the algorithm.**
>
> The key superiority w.r.t. the prior related work in OT is that our method works for general cost functionals $\mathcal{F}$, allowing to approach previously unreleased problems. More precisely, in the task of class-guided dataset transfer (the key testbed of this paper), our method significantly outperforms prior OT alternatives. We also show that for the dataset transfer, we work comparably or better than existing non-OT-based alternatives, see Tables 1, 2.
>
> **Concluding remarks.** Please respond to our post to let us know if the clarifications above suitably address your concerns about our work. We are happy to address any remaining points during the discussion phase; if the responses above are sufficient, we kindly ask that you consider raising your score.
>
> **References**
>
> - Alexander Korotin, Lingxiao Li, Aude Genevay, Justin Solomon, Alexander Filippov, and Evgeny Burnaev. Do neural optimal transport solvers work? a continuous wasserstein-2 benchmark. Advances in Neural Information Processing Systems, 2021b.
>
> - Jiaojiao Fan, Qinsheng Zhang, Amirhossein Taghvaei, Yongxin Chen: Variational Wasserstein gradient flow. https://arxiv.org/abs/2112.02424 (ICML 2022).
>
> - Makkuva, A., Taghvaei, A., Oh, S., \& Lee, J. (2020, November). Optimal transport mapping via input convex neural networks. In International Conference on Machine Learning (pp. 6672-6681). PMLR.
>
> - Taghvaei, Amirhossein and Jalali, Amin. 2-{W}asserstein Approximation via Restricted Convex Potentials with Application to Improved Training for {GAN}s. arXiv preprint arXiv:1902.07197. 2019.

---

### Official Review · Reviewer_7wv1 · 2022-10-24

**Confidence:** 3
**Correctness:** 4
**Technical Novelty And Significance:** 3
**Empirical Novelty And Significance:** 2
**Recommendation:** 6

**Clarity, Quality, Novelty And Reproducibility:**

The language of the paper is clear and numerical experiments seem reproducible.

**Strength And Weaknesses:**

Strength:
- the general ideas considered in the paper are novel and interesting.
- the application that considered in the paper is interesting.
- the mathematical results seem correct

Weakness:
- the discussion of related works does not go deep and it is merely a list.
- there are many results presented that makes reading the paper a bit difficult. It might not be necessarily relevant to the main objectives. I think it is better to reduce them and expand the main ones, adding explanations and clarifications.




**Summary Of The Paper:**

The paper is addressing the problem of finding optimal transport plans for general cost functionals. The algorithms is based on a saddle point reformulation of the problem where the marginal constraint is formulated as maximization. The plan on (x,y) is represented as a stochastic map from (x,z) to  (x,T(z,x)) where z is the latent random variable and T(z,x) represents the conditional distribution p_{y|x}, in a similar style as conditional GANs. The application of the proposed algorithm to a data transfer problem with class preservation is described and the results are illustrated.

**Summary Of The Review:**

I think the paper has enough content to be a good contribution but It should be presented in a way that highlights the main contributions and a more deeper discussion of the related works.

---

> ### Author Response · Authors · 2022-11-13
> **Answers to the Reviewer 7wv1**
>
> Thank you for spending time reviewing our paper and providing useful feedback that will help us improve the manuscript. Please find the answers to your questions below.
>
> **Q1: The discussion of related works does not go deep and it is merely a list.  Deeper discussion of the related works.**
>
> We extended the discussion in the related work section. In particular, we added more details about domain adaptation methods and discrete OT solvers. Please consider the revised version.
>
> **Q2: here are many results presented that makes reading the paper a bit difficult. It might not be necessarily relevant to the main objectives. I think it is better to reduce them and expand the main ones, adding explanations and clarifications. It should be presented in a way that highlights the main contributions**
>
> All of the theoretical results provided in Sections 3 and 4 are novel and form our contribution. In Subsections 3.1 and 3.2, we theoretically derive the objective to learn OT plans for general cost functionals $\mathcal{F}$. In Section 3.3, we detail how the objective can be optimized in practice with neural networks. In Section 4, we propose and justify the particular functional $\mathcal{F}_{G}$ for the dataset transfer task. In turn, Section 6 illustrates the capabilities of our method in various setups.
>
> In our view, removing any of those results might harm readability. Could you please suggest the results that, in your opinion, make reading the paper a bit difficult?
>
> **Concluding remarks.** Please respond to our post to let us know if the clarifications above suitably address your concerns about our work. We are happy to address any remaining points during the discussion phase.

---

### Official Review · Reviewer_MBuL · 2022-10-27

**Confidence:** 3
**Correctness:** 4
**Technical Novelty And Significance:** 4
**Empirical Novelty And Significance:** 4
**Recommendation:** 5

**Clarity, Quality, Novelty And Reproducibility:**

For the details of these points, please see my review of "Strength and Weakness". However, to give a high-level answer:

**Clarity:** The paper is written very clearly. It is easy to follow. The theoretical contributions are clearly stated and the limitations are honestly discussed.

**Quality:** The paper has high quality. The experiments, visualizations, and results are thorough. The proofs look correct to me.

**Novelty:** The paper has novelty, and can guide the community toward a very modern adaptation of OT thanks to Section 4.

**Reproducibility:** The paper provides all the source codes, and the main algorithm is very clear. I cannot foresee any issues with reproducibility. The limitations are also discussed.

**Strength And Weaknesses:**

**Strength:** Firstly, I would like to thank the author for such a clear motivation, literature review, theoretical analysis, numerical experiments, implementation, and discussions. I am **not** an expert in OT, but I used it in my research a couple of times, and as far as I am concerned the paper looks like a great fit for the community.

**Weaknesses:**
*I list some minor weaknesses. I am going to stay active during the discussion period in case the authors update the document or have questions about my review.*

***Minor Weaknesses/Questions***
- Section 1 starts very 'quick'. What is "unpaired restoration and style translation", for example?
- Section 2: In the beginning we have $\mathcal{X} = \mathcal{Y}$. Is this a typo?
- For Wasserstein-1 and Wasserstein-2 terminologies in Section 2, the given citations are specifically recent NN-based papers. However, the terminology definition comes from much earlier if I am not wrong.
- "General OT" (5) requires citations. It may look like this is a new definition.
- Section 3.1: "which alternative to (6)" -> typo
- Theorem 1: "minimax" -> maximin ? (The rest says maximin)
- Question: Is (11) used in any other paper before? If not, Proposition 1 can be a theorem.
- Section 6: "as well as the trained networks" -> "along with" is maybe better?
- Section 3.3 looks shorter than it has to be, in my view. There is a huge relevant literature in Stochastic Programming, but here it is a little hard to relate. The proposed unbiased estimator comes without discussion.

**UPDATE**
After reading (Korotin et al., 2022c) I am less enthusiastic about this work. I like the paper, but in the presence of the other one, I think the setting taken here can be derived relatively easily. I would like to update my score to "5" instead of "8".

**Summary Of The Paper:**

The authors first take a generalization to the strong and weak optimal transport problems by replacing the objective function from an expected cost wrt the transport plan to a general lower-semicontinuous functional. This formulation coincides with the regularized optimal transport, however, it is easier to adjust under different domains as one can directly design the functional rather than the additive regularizer.  It is shown that this problem admits a strong dual which is a max-min optimization problem, resembling the Kantorovich dual. The authors then show how to retrieve the optimal transportation plan from this dual, which coincides with the saddle points. By exploiting this result, they derive an NN-based solution algorithm, and afterward, they show how one can benefit from such a framework in a case where the underlying domain is concerned with having a "cost" representation that is beyond distances. They illustrate the performance of these methods over various numerical experiment settings.

**Summary Of The Review:**

This is one of the best papers I have reviewed in a long time. From the accuracy of the literature review to the solid theoretical results, useful reformulations, interesting experiments, and honest but very efficient solution algorithms, the paper is written extremely well.

---

> ### Author Response · Authors · 2022-11-13
> **Answers to the Reviewer MBuL**
>
> Thank you very much for your detailed analysis of our paper and thoughtful suggestions on paper improvements. We took your feedback onboard and uploaded a revised version of our paper. We provide a response to your comments below.
>
> **Q1: Section 1 starts very 'quick'. What is "unpaired restoration and style translation", for example?**
>
> Thank you for bringing this to our attention. We added an early reference to papers
> that describe the style translation (Zhu et al., 2017, Figures 1, 2) and image restoration (Lugmayr et al., 2020) setups.
>
> **Q2: Section 2: In the beginning we have $\mathcal{X}=\mathcal{Y}$. Is this a typo?**
>
> No, this is not a typo. In our paper, we consider the same $\mathbb{R}^D$ space for $\mathcal{X}$ and $\mathcal{Y}$. This is needed to apply the theoretical findings of (Paty and Cuturi, 2020) in our paper.
> Note that the assumption $\mathcal{X}=\mathcal{Y}$ is not a limitation. Indeed, in case $\mathcal{X}\subset \mathbb{R}^{D_{1}}$ and $\mathcal{Y}\subset \mathbb{R}^{D_{2}}$ with $D_{1}\neq D_{2}$, one may inject both spaces to the same space $\mathcal{X}'=\mathcal{Y}'=\mathbb{R}^{\max(D_{1},D_{2})}$.
>
>
> **Q3: For Wasserstein-1 and Wasserstein-2 terminology definition comes from much earlier.**
>
> Following your comment, we replaced the current references with references to the fundamental OT literature: (Villani, 2008, Paragraph 1) and (Santambrogio, 2015. Paragraphs 1, 2).
>
>
> **Q4: "General OT" (5) requires citations. It may look like this is a new definition.**
>
> We agree. In the revision, we added a reference to the OT formulation proposed in (Paty and Cuturi, 2020).
>
> **Q5: Typos and minor edits.**
>
> Thanks for carefully checking the paper. We fixed all the typos and renamed Proposition 1 to Theorem 3.
>
> **Q6: Section 3.3 looks shorter than it has to be, in my view. There is a huge relevant literature in Stochastic Programming, but here it is a little hard to relate. The proposed unbiased estimator comes without discussion**
>
> Thank you for your suggestion. We added the details on the used optimizer (Adam) to this section. Could you please recommend which additional discussions or literature on stochastic programming we should add?
>
> **Concluding remarks.** Please respond to our post to let us know if the clarifications above suitably address your concerns about our work.
>
> **References**
>
> - François-Pierre Paty and Marco Cuturi. Regularized optimal transport is ground cost adversarial. InInternational Conference on Machine Learning, pp. 7532–7542. PMLR, 2020
>
> - Jun-Yan Zhu, Taesung Park, Phillip Isola, and Alexei A Efros. Unpaired image-to-image translation using cycle-consistent adversarial networks. In Proceedings of the IEEE international conference on computer vision, pp. 2223–2232, 2017
>
> - Lugmayr, A., Danelljan, M., and Timofte, R. (2020). Ntire 2020 challenge on real-world image super-resolution: Methods and results. In Proceedings of the IEEE/CVF Conference on Computer Vision and Pattern Recognition Workshops (pp. 494-495).
>
> - Cédric Villani. Optimal transport: old and new, volume 338. Springer Science  Business Media
>
> - Filippo Santambrogio. Optimal transport for applied mathematicians. Birkäuser, NY, 55(58-63):94, 2015

---

> > ### Comment · Reviewer_MBuL · 2022-11-16
> > **Acknowledging the response**
> >
> > Dear Authors,
> >
> > Thank you very much for your reply. This is great -- I can see most of my comments are addressed, and the document is updated accordingly.
> >
> > Quick note: I don't have a strong preference for the "Theorem 3" name. You can feel free to change it back to Prop. I just wanted to say the result looks interesting enough to be named a Thm.
> >
> > A minor typo: From the newly added sentences "and our Algorithm 1 for general OT 5 [...]" should it be OT (5)?
> >
> > Great to see more discussion in S3.3. However, Adam is just *an* optimization algorithm. What I meant was that in stochastic programming there are also lots of different techniques to solve similar problems, which are not discussed. The proposed algorithm resembles the SAA methods, but there are many more. I was just curious to know how this algorithm relates; because variants of this problem frequently appear in the stochastic programming literature.
> >
> > Finally, I realized that (Korotin et al., 2022c) is also a paper that is under review, and I don't know how much we can rely on the comparisons with this paper. I will quickly check some of these papers, and come back if I have further questions. For now, things look good, thank you for your time!

---

### Author Response · Authors · 2022-11-13
**Revised Paper**

We appreciate the reviewers' thoughtful feedback. We are excited that the reviewers find our ideas novel, interesting, and well-motivated (MBuL, 7wv1, AhFA), and our paper clear and easy to follow (MBuL, AhFA, 7wv1). We also appreciate the fact that reviewers find our application interesting (7wv1) and with clear and reproducible numerical experiments (MBuL, 7wv1, tFkj, AhFA).


Please consider the updated paper and appendices. The edits are highlighted by the **blue** color in the revised version of the submission. The main edits are listed below:

- Sections 1 and 2 now have additional references. All typos have been corrected **(MBuL)**.

- We enhanced Section 5 (related work) with more in-depth discussions **(7wv1)**.

- In Section 3, we discussed the difference from the (Korotin et al., 2022b) in more details **(AhFA)**.

We would be grateful if we could hear your feedback regarding our answers to the reviews and our revised submission. We are happy to address any remaining points during the remaining period.

---

> ### Author Response · Authors · 2022-11-17
> **Revised Paper**
>
> Dear reviewers, we updated the paper and appendices.
>
> - We added a comparison to the ICNN-based OT method(Makkuva et al., 2019; Korotin et al., 2021) in Appendix-F. We considered both **related** and **unrelated** dataset transfer problems in this experiment **(tFkj)**.
>
> - In Appendix-G, we evaluated our deterministic transport map $T$ with cost function $\mathcal{F}_{G}$ on a dataset transfer problem with non-default class correspondence between domains **(AhFA)**.
>
> We are happy to address any additional questions during the remaining period.

---

### Decision · Program_Chairs · 2023-01-20

**Decision:**

Reject

**Justification For Why Not Higher Score:**

This paper is incremental compared to Korotin 2022c. indeed, it is more general but the related properties and algorithm can be easily derived from that paper.

**Justification For Why Not Lower Score:**

NA

**Metareview: Summary, Strengths And Weaknesses:**

The paper proposes a framework and an algorithm for solving a so-called general OT problem.

While reviewers agree that the paper provides a solution to a new problem, after the live discussion, we agreed that the contributions are incremental compared to Korotin 2022c and other (strongly related) papers submitted to ICLR, especially from the theoretical perspective. When we look at the practical contributions (on class-guided NOT for instance), we also concur on the fact that the addressed problem (with partial label on target domain) is not a strong contribution and it has been only validated on toy(ish) MNIST problems.
Based on these points, we can not recommend acceptance.

**Summary Of Ac-Reviewer Meeting:**

We all agreed that the contributions were incremental compared to Korotin 2022c and other papers submitted to ICLR and the new application (class-guided OT) is not strong enough